# Trust Region Policy Optimisation in Multi-Agent Reinforcement Learning

**Jakub Grudzien Kuba**[1,2*], **Ruiqing Chen**[3,*], **Muning Wen**[4], **Ying Wen**[4],
**Fanglei Sun**[3], **Jun Wang**[5], **Yaodong Yang**[6,†]
[1]University of Oxford, [2]Huawei R&D UK, [3]ShanghaiTech University,
[4]Shanghai Jiao Tong University [5]University College London
[6]Institute for AI, Peking University & BIGAI
[†]Corresponding to: yaodong.yang@pku.edu.cn

## Abstract

Trust region methods rigorously enabled reinforcement learning (RL) agents to learn monotonically improving policies, leading to superior performance on a variety of tasks. Unfortunately, when it comes to multi-agent reinforcement learning (MARL), the property of monotonic improvement may not simply apply; this is because agents, even in cooperative games, could have conflicting directions of policy updates. As a result, achieving a guaranteed improvement on the joint policy where each agent acts individually remains an open challenge. In this paper, we extend the theory of trust region learning to cooperative MARL. Central to our findings are the *multi-agent advantage decomposition lemma* and the *sequential policy update scheme*. Based on these, we develop *Heterogeneous-Agent Trust Region Policy Optimisation* (HATPRO) and *Heterogeneous-Agent Proximal Policy Optimisation* (HAPPO) algorithms. Unlike many existing MARL algorithms, HATRPO/HAPPO do not need agents to share parameters, nor do they need any restrictive assumptions on decomposibility of the joint value function. Most importantly, we justify in theory the monotonic improvement property of HATRPO/HAPPO. We evaluate the proposed methods on a series of Multi-Agent MuJoCo and StarCraftII tasks. Results show that HATRPO and HAPPO significantly outperform strong baselines such as IPPO, MAPPO and MADDPG on all tested tasks, thereby establishing a new state of the art.

## 1 Introduction

Policy gradient (PG) methods have played a major role in recent developments of reinforcement learning (RL) algorithms (Silver et al., 2014; Schulman et al., 2015a; Haarnoja et al., 2018). Among the many PG variants, *trust region learning* (Kakade & Langford, 2002), with two typical embodiments of *Trust Region Policy Optimisation* (TRPO) (Schulman et al., 2015a) and *Proximal Policy Optimisation* (PPO) (Schulman et al., 2017) algorithms, offer supreme empirical performance in both discrete and continuous RL problems (Duan et al., 2016; Mahmood et al., 2018). The effectiveness of trust region methods largely stems from their theoretically-justified policy iteration procedure. By optimising the policy within a *trustable* neighbourhood of the current policy, thus avoiding making aggressive updates towards risky directions, trust region learning enjoys the guarantee of monotonic performance improvement at every iteration.

In multi-agent reinforcement learning (MARL) settings (Yang & Wang, 2020), naively applying policy gradient methods by considering other agents as a part of the environment can lose its effectiveness. This is intuitively clear: once a learning agent updates its policy, so do its opponents; this however changes the loss landscape of the learning agent, thus harming the improvement effect from the PG update. As a result, applying independent PG updates in MARL offers poor convergence property (Claus & Boutilier, 1998). To address this, a learning paradigm named *centralised training with decentralised execution* (CTDE) (Lowe et al., 2017b; Foerster et al., 2018; Zhou et al., 2021) was developed. In CTDE, each agent is equipped with a joint value function which, during

---

[*]First two authors contribute equally. Code is available at https://github.com/PKU-MARL/TRPO-PPO-in-MARL.

training, has access to the global state and opponents' actions. With the help of the centralised value function that accounts for the non-stationarity caused by others, each agent adapts its policy parameters accordingly. As such, the CTDE paradigm allows a straightforward extension of single-agent PG theorems (Sutton et al., 2000; Silver et al., 2014) to multi-agent scenarios (Lowe et al., 2017b; Kuba et al., 2021; Mguni et al., 2021). Consequently, fruitful multi-agent policy gradient algorithms have been developed (Foerster et al., 2018; Peng et al., 2017; Zhang et al., 2020; Wen et al., 2018; 2020; Yang et al., 2018).

Unfortunately, existing CTDE methods offer no solution of how to perform trust region learning in MARL. Lack of such an extension impedes agents from learning monotonically improving policies in a stable manner. Recent attempts such as IPPO (de Witt et al., 2020a) and MAPPO (Yu et al., 2021) have been proposed to fill such a gap; however, these methods are designed for agents that are *homogeneous* (i.e., sharing the same action space and policy parameters), which largely limits their applicability and potentially harm the performance. As we show in Proposition 1 later, parameter sharing could suffer from an exponentially-worse suboptimal outcome. On the other hand, although IPPO/MAPPO can be practically applied in a non-parameter sharing way, it still lacks the essential theoretical property of trust region learning, which is the monotonic improvement guarantee.

In this paper, we propose the first theoretically-justified trust region learning framework in MARL. The key to our findings are the *multi-agent advantage decomposition lemma* and the *sequential policy update scheme*. With the advantage decomposition lemma, we introduce a multi-agent policy iteration procedure that enjoys the monotonic improvement guarantee. To implement such a procedure, we propose two practical algorithms: *Heterogeneous-Agent Trust Region Policy Optimisation* (HATRPO) and *Heterogeneous-Agent Proximal Policy Optimisation* (HAPPO). HATRPO/HAPPO adopts the sequential update scheme, which saves the cost of maintaining a centralised critic for each agent in CTDE. Importantly, HATRPO/HAPPO does not require homogeneity of agents, nor any other restrictive assumptions on the decomposibility of the joint Q-function (Rashid et al., 2018). We evaluate HATRPO and HAPPO on benchmarks of StarCraftII and Multi-Agent MuJoCo against strong baselines such as MADDPG (Lowe et al., 2017a), IPPO (de Witt et al., 2020b) and MAPPO (Yu et al., 2021); results clearly demonstrate its state-of-the-art performance across all tested tasks.

## 2 PRELIMINARIES

In this section, we first introduce problem formulation and notations for MARL, and then briefly review trust region learning in RL and discuss the difficulty of extending it to MARL. We end by surveying existing MARL work that relates to trust region methods and show their limitations.

### 2.1 COOPERATIVE MARL PROBLEM FORMULATION AND NOTATIONS

We consider a Markov game (Littman, 1994), which is defined by a tuple $\langle \mathcal{N}, \mathcal{S}, \mathcal{A}, P, r, \gamma \rangle$. Here, $\mathcal{N} = \{1, \ldots, n\}$ denotes the set of agents, $\mathcal{S}$ is the finite state space, $\mathcal{A} = \prod_{i=1}^{n} \mathcal{A}^i$ is the product of finite action spaces of all agents, known as the joint action space, $P : \mathcal{S} \times \mathcal{A} \times \mathcal{S} \to [0, 1]$ is the transition probability function, $r : \mathcal{S} \times \mathcal{A} \to \mathbb{R}$ is the reward function, and $\gamma \in [0, 1)$ is the discount factor. The agents interact with the environment according to the following protocol: at time step $t \in \mathbb{N}$, the agents are at state $\mathrm{s}_t \in \mathcal{S}$; every agent $i$ takes an action $\mathrm{a}_t^i \in \mathcal{A}^i$, drawn from its policy $\pi^i(\cdot|\mathrm{s}_t)$, which together with other agents' actions gives a joint action $\mathbf{a}_t = (\mathrm{a}_t^1, \ldots, \mathrm{a}_t^n) \in \mathcal{A}$, drawn from the joint policy $\boldsymbol{\pi}(\cdot|\mathrm{s}_t) = \prod_{i=1}^{n} \pi^i(\cdot^i|\mathrm{s}_t)$; the agents receive a joint reward $\mathrm{r}_t = r(\mathrm{s}_t, \mathbf{a}_t) \in \mathbb{R}$, and move to a state $\mathrm{s}_{t+1}$ with probability $P(\mathrm{s}_{t+1}|\mathrm{s}_t, \mathbf{a}_t)$. The joint policy $\boldsymbol{\pi}$, the transition probabililty function $P$, and the initial state distribution $\rho^0$, induce a marginal state distribution at time $t$, denoted by $\rho_{\boldsymbol{\pi}}^t$. We define an (improper) marginal state distribution $\rho_{\boldsymbol{\pi}} \triangleq \sum_{t=0}^{\infty} \gamma^t \rho_{\boldsymbol{\pi}}^t$. The state value function and the state-action value function are defined: $V_{\boldsymbol{\pi}}(s) \triangleq \mathbb{E}_{\mathbf{a}_{0:\infty} \sim \boldsymbol{\pi}, \mathrm{s}_{1:\infty} \sim P}\left[\sum_{t=0}^{\infty} \gamma^t \mathrm{r}_t \big| \mathrm{s}_0 = s\right]$ and $Q_{\boldsymbol{\pi}}(s, \boldsymbol{a}) \triangleq \mathbb{E}_{\mathrm{s}_{1:\infty} \sim P, \mathbf{a}_{1:\infty} \sim \boldsymbol{\pi}}\left[\sum_{t=0}^{\infty} \gamma^t \mathrm{r}_t \big| \mathrm{s}_0 = s, \mathbf{a}_0 = \boldsymbol{a}\right]$. The advantage function is written as $A_{\boldsymbol{\pi}}(s, \boldsymbol{a}) \triangleq Q_{\boldsymbol{\pi}}(s, \boldsymbol{a}) - V_{\boldsymbol{\pi}}(s)$. In this paper, we consider a *fully-cooperative* setting where all agents share the same reward function, aiming to maximise the expected total reward:

$$J(\boldsymbol{\pi}) \triangleq \mathbb{E}_{\mathrm{s}_{0:\infty} \sim \rho_{\boldsymbol{\pi}}^{0:\infty}, \mathbf{a}_{0:\infty} \sim \boldsymbol{\pi}}\left[\sum_{t=0}^{\infty} \gamma^t \mathrm{r}_t\right].$$

Throughout this paper, we pay close attention to the contribution to performance from different subsets of agents. Before proceeding to our methods, we introduce following novel definitions.

**Definition 1.** *Let $i_{1:m}$ denote an ordered subset $\{i_1, \ldots, i_m\}$ of $\mathcal{N}$, and let $-i_{1:m}$ refer to its complement. We write $i_k$ when we refer to the $k^{th}$ agent in the ordered subset. Correspondingly, the multi-agent state-action value function is defined as*

$$Q_{\boldsymbol{\pi}}^{i_{1:m}}\left(s, \boldsymbol{a}^{i_{1:m}}\right) \triangleq \mathbb{E}_{\mathbf{a}^{-i_{1:m}} \sim \boldsymbol{\pi}^{-i_{1:m}}}\left[Q_{\boldsymbol{\pi}}\left(s, \boldsymbol{a}^{i_{1:m}}, \mathbf{a}^{-i_{1:m}}\right)\right],$$

*and for disjoint sets $j_{1:k}$ and $i_{1:m}$, the* multi-agent advantage function *is*

$$A_{\boldsymbol{\pi}}^{i_{1:m}}\left(s, \boldsymbol{a}^{j_{1:k}}, \boldsymbol{a}^{i_{1:m}}\right) \triangleq Q_{\boldsymbol{\pi}}^{j_{1:k}, i_{1:m}}\left(s, \boldsymbol{a}^{j_{1:k}}, \boldsymbol{a}^{i_{1:m}}\right) - Q_{\boldsymbol{\pi}}^{j_{1:k}}\left(s, \boldsymbol{a}^{j_{1:k}}\right). \tag{1}$$

Hereafter, the joint policies $\boldsymbol{\pi} = (\pi^1, \ldots, \pi^n)$ and $\bar{\boldsymbol{\pi}} = (\bar{\pi}^1, \ldots, \bar{\pi}^n)$ shall be thought of as the "current", and the "new" joint policy that agents update towards, respectively.

## 2.2 TRUST REGION ALGORITHMS IN REINFORCEMENT LEARNING

Trust region methods such as TRPO (Schulman et al., 2015a) were proposed in single-agent RL with an aim of achieving a monotonic improvement of $J(\pi)$ at each iteration. Formally, it can be described by the following theorem.

**Theorem 1.** *(Schulman et al., 2015a, Theorem 1) Let $\pi$ be the current policy and $\bar{\pi}$ be the next candidate policy. We define $L_\pi(\bar{\pi}) = J(\pi) + \mathbb{E}_{s \sim \rho_\pi, a \sim \bar{\pi}}[A_\pi(s, a)], \mathrm{D}_{KL}^{max}(\pi, \bar{\pi}) = \max_s \mathrm{D}_{KL}(\pi(\cdot|s), \bar{\pi}(\cdot|s))$. Then the inequality of*

$$J(\bar{\pi}) \geq L_\pi(\bar{\pi}) - C\mathrm{D}_{KL}^{max}(\pi, \bar{\pi}) \tag{2}$$

*holds, where $C = \frac{4\gamma \max_{s,a} |A_\pi(s,a)|}{(1-\gamma)^2}$.*

The above theorem states that as the distance between the current policy $\pi$ and a candidate policy $\bar{\pi}$ decreases, the surrogate $L_\pi(\bar{\pi})$, which involves only the current policy's state distribution, becomes an increasingly accurate estimate of the actual performance metric $J(\bar{\pi})$. Based on this theorem, an iterative trust region algorithm is derived; at iteration $k+1$, the agent updates its policy by

$$\pi_{k+1} = \arg\max_\pi \left(L_{\pi_k}(\pi) - C\mathrm{D}_{KL}^{max}(\pi_k, \pi)\right).$$

Such an update guarantees a monotonic improvement of the policy, i.e., $J(\pi_{k+1}) \geq J(\pi_k)$. To implement this procedure in practical settings with parameterised policies $\pi_\theta$, Schulman et al. (2015a) approximated the KL-penalty with a KL-constraint, which gave rise to the TRPO update of

$$\theta_{k+1} = \arg\max_\theta L_{\pi_{\theta_k}}(\pi_\theta), \quad \text{subject to } \mathbb{E}_{s \sim \rho_{\pi_{\theta_k}}}[\mathrm{D}_{KL}(\pi_{\theta_k}, \pi_\theta)] \leq \delta. \tag{3}$$

At each iteration $k+1$, TRPO constructs a KL-ball $\mathfrak{B}_\delta(\pi_{\theta_k})$ around the policy $\pi_{\theta_k}$, and optimises $\pi_{\theta_{k+1}} \in \mathfrak{B}_\delta(\pi_{\theta_k})$ to maximise $L_{\pi_{\theta_k}}(\pi_\theta)$. By Theorem 1, we know that the surrogate objective $L_{\pi_{\theta_k}}(\pi_\theta)$ is close to the true reward $J(\pi_\theta)$ within $\mathfrak{B}_\delta(\pi_{\theta_k})$; therefore, $\pi_{\theta_k}$ leads to improvement. Furthermore, to save the cost on $\mathbb{E}_{s \sim \rho_{\pi_{\theta_k}}}[\mathrm{D}_{KL}(\pi_{\theta_k}, \pi_\theta)]$ when computing Equation (3), Schulman et al. (2017) proposed an approximation solution to TRPO that uses only first order derivatives, known as PPO. PPO optimises the policy parameter $\theta_{k+1}$ by maximising the *PPO-clip* objective of

$$L_{\pi_{\theta_k}}^{\text{PPO}}(\pi_\theta) = \mathbb{E}_{s \sim \rho_{\pi_{\theta_k}}, a \sim \pi_{\theta_k}}\left[\min\left(\frac{\pi_\theta(a|s)}{\pi_{\theta_k}(a|s)}A_{\pi_{\theta_k}}(s, a), \mathrm{clip}\left(\frac{\pi_\theta(a|s)}{\pi_{\theta_k}(a|s)}, 1 \pm \epsilon\right)A_{\pi_{\theta_k}}(s, a)\right)\right]. \tag{4}$$

The *clip* operator replaces the ratio $\frac{\pi_\theta(a|s)}{\pi_{\theta_k}(a|s)}$ with $1 + \epsilon$ or $1 - \epsilon$, depending on whether or not the ratio is beyond the threshold interval. This effectively enables PPO to control the size of policy updates.

## 2.3 LIMITATIONS OF EXISTING TRUST REGION METHODS IN MARL

Extending trust region methods to MARL is highly non-trivial. One naive approach is to equip all agents with one shared set of parameters and use agents' aggregated trajectories to conduct policy optimisation at every iteration. This approach was adopted by MAPPO (Yu et al., 2021) in which the policy parameter $\theta$ is optimised by maximising the objective of

$$L_{\boldsymbol{\pi}_{\theta_k}}^{\text{MAPPO}}(\pi_\theta) = \sum_{i=1}^n \mathbb{E}_{s \sim \rho_{\boldsymbol{\pi}_{\theta_k}}, \mathbf{a} \sim \boldsymbol{\pi}_{\theta_k}}\left[\min\left(\frac{\pi_\theta(a^i|s)}{\pi_{\theta_k}(a^i|s)}A_{\boldsymbol{\pi}_{\theta_k}}(s, \mathbf{a}), \mathrm{clip}\left(\frac{\pi_\theta(a^i|s)}{\pi_{\theta_k}(a^i|s)}, 1 \pm \epsilon\right)A_{\boldsymbol{\pi}_{\theta_k}}(s, \mathbf{a})\right)\right]. \tag{5}$$

Unfortunately, this simple approach has significant drawbacks. An obvious demerit is that parameter sharing requires that all agents have identical action spaces, *i.e.*, $\mathcal{A}^i = \mathcal{A}^j, \forall i, j \in \mathcal{N}$, which limits the class of MARL problems to solve. Importantly, enforcing parameter sharing is equivalent to putting a constraint $\theta^i = \theta^j, \forall i, j \in \mathcal{N}$ on the joint policy space. In principle, this can lead to a suboptimal solution. To elaborate, we demonstrate through an example in the following proposition.

**Proposition 1.** *Let's consider a fully-cooperative game with an even number of agents $n$, one state, and the joint action space $\{0, 1\}^n$, where the reward is given by $r(\mathbf{0}^{n/2}, \mathbf{1}^{n/2}) = r(\mathbf{1}^{n/2}, \mathbf{0}^{n/2}) = 1$, and $r(\mathbf{a}^{1:n}) = 0$ for all other joint actions. Let $J^*$ be the optimal joint reward, and $J^*_{share}$ be the optimal joint reward under the shared policy constraint. Then*

$$\frac{J^*_{share}}{J^*} = \frac{2}{2^n}.$$

For proof see Appendix B. In the above example, we show that parameter sharing can lead to a suboptimal outcome that is exponentially worse with the increasing number of agents. We also provide an empirical verification of this proposition in Appendix F.

Apart from parameter sharing, a more general approach to extend trust region methods in MARL is to endow all agents with their own parameters, and at each iteration $k + 1$, agents construct trust regions of $\{\mathfrak{B}_\delta(\pi^i_{\theta^i_k})\}_{i \in \mathcal{N}}$, and optimise their objectives $\{L_{\boldsymbol{\pi}_{\boldsymbol{\theta}_k}}(\pi^i_{\theta^i} \boldsymbol{\pi}^{-i}_{\boldsymbol{\theta}^{-i}_k})\}_{i \in \mathcal{N}}$.

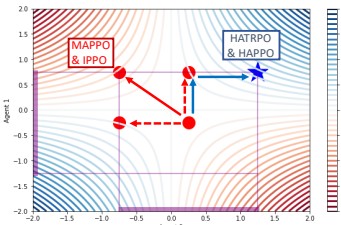

Figure 1: Example of a two-player differentiable game with $r(a^1, a^2) = a^1 a^2$. We initialise two Gaussian policies with $\mu^1 = -0.25$, $\mu^2 = 0.25$. The purple intervals represent the KL-ball of $\delta = 0.5$. Individual trust region updates (red) decrease the joint return, whereas our sequential update (blue) leads to improvement.

Admittedly, this approach can still be supported by the current MAPPO implementation (Yu et al., 2021) if one turns off parameter sharing, thus distributing the summation in Equation (5) to all agents. However, such an approach cannot offer a rigorous guarantee of monotonic improvement during training. In fact, agents' local improvements in performance can jointly lead to a worse outcome. For example, in Figure 1, we design a single-state differential game where two agents draw their actions from Gaussian distributions with learnable means $\mu^1$, $\mu^2$ and unit variance, and the reward function is $r(a^1, a^2) = a^1 a^2$. The failure of MAPPO-style approach comes from the fact that, although the reward function increases in each of the agents' (one-dimensional) update directions, it decreases in the joint (two-dimensional) update direction.

Having seen the limitations of existing trust region methods in MARL, in the following sections, we first introduce a multi-agent policy iteration procedure that enjoys theoretically-justified monotonic improvement guarantee. To implement this procedure, we propose *HATRPO* and *HAPPO* algorithms, which offer practical solutions to apply trust region learning in MARL without the necessity of assuming homogeneous agents while still maintaining the monotonic improvement property.

## 3 MULTI-AGENT TRUST REGION LEARNING

The purpose of this section is to develop a theoretically-justified trust region learning procedure in the context of multi-agent learning. In Subsection 3.1, we present the policy iteration procedure with monotonic improvement guarantee, and in Subsection 3.2, we analyse its properties during training and at convergence. Throughout the work, we make the following regularity assumptions.

**Assumption 1.** *There exists $\eta \in \mathbb{R}$, such that $0 < \eta \ll 1$, and for every agent $i \in \mathcal{N}$, the policy space $\Pi^i$ is $\eta$-soft; that means that for every $\pi^i \in \Pi^i$, $s \in \mathcal{S}$, and $a^i \in \mathcal{A}^i$, we have $\pi^i(a^i|s) \geq \eta$.*

### 3.1 TRUST REGION LEARNING IN MARL WITH MONOTONIC IMPROVEMENT GUARANTEE

We start by introducing a pivotal lemma which shows that the joint advantage function can be decomposed into a summation of each agent's local advantages. Importantly, this lemma offers a critical intuition behind the sequential policy-update scheme that our algorithms later apply.

**Lemma 1** (Multi-Agent Advantage Decomposition). *In any cooperative Markov games, given a joint policy $\boldsymbol{\pi}$, for any state $s$, and any agent subset $i_{1:m}$, the below equations holds.*

$$A^{i_{1:m}}_{\boldsymbol{\pi}}\left(s, \mathbf{a}^{i_{1:m}}\right) = \sum_{j=1}^m A^{i_j}_{\boldsymbol{\pi}}\left(s, \mathbf{a}^{i_{1:j-1}}, a^{i_j}\right).$$

For proof see Appendix C.2. Notably, Lemma 1 holds in general for cooperative Markov games, with no need for any assumptions on the decomposibility of the joint value function such as those in VDN (Sunehag et al., 2018), QMIX (Rashid et al., 2018) or Q-DPP (Yang et al., 2020).

Lemma 1 indicates an effective approach to search for the direction of performance improvement (i.e., joint actions with positive advantage values) in multi-agent learning. Specifically, let agents take actions sequentially by following an arbitrary order $i_{1:n}$, assuming agent $i_1$ takes an action $\bar{a}^{i_1}$ such that $A^{i_1}(s, \bar{a}^{i_1}) > 0$, and then, for the rest $m = 2, \ldots, n$, the agent $i_m$ takes an action $\bar{a}^{i_m}$ such that $A^{i_m}(s, \bar{a}^{i_{1:m-1}}, \bar{a}^{i_m}) > 0$. For the induced joint action $\bar{a}$, Lemma 1 assures that $A_{\pi_\theta}(s, \bar{a})$ is positive, thus the performance is guaranteed to improve. To formally extend the above process into a policy iteration procedure with monotonic improvement guarantee, we need the following definitions.

**Definition 2.** *Let $\pi$ be a joint policy, $\bar{\pi}^{i_{1:m-1}} = \prod_{j=1}^{m-1} \bar{\pi}^{i_j}$ be some **other** joint policy of agents $i_{1:m-1}$, and $\hat{\pi}^{i_m}$ be some **other** policy of agent $i_m$. Then*

$$L_{\pi}^{i_{1:m}}\left(\bar{\pi}^{i_{1:m-1}}, \hat{\pi}^{i_m}\right) \triangleq \mathbb{E}_{s \sim \rho_{\pi}, \mathbf{a}^{i_{1:m-1}} \sim \bar{\pi}^{i_{1:m-1}}, a^{i_m} \sim \hat{\pi}^{i_m}}\left[A_{\pi}^{i_m}\left(s, \mathbf{a}^{i_{1:m-1}}, a^{i_m}\right)\right].$$

Note that, for any $\bar{\pi}^{i_{1:m-1}}$, we have

$$L_{\pi}^{i_{1:m}}\left(\bar{\pi}^{i_{1:m-1}}, \pi^{i_m}\right) = \mathbb{E}_{s \sim \rho_{\pi}, \mathbf{a}^{i_{1:m-1}} \sim \bar{\pi}^{i_{1:m-1}}, a^{i_m} \sim \pi^{i_m}}\left[A_{\pi}^{i_m}\left(s, \mathbf{a}^{i_{1:m-1}}, a^{i_m}\right)\right]$$
$$= \mathbb{E}_{s \sim \rho_{\pi}, \mathbf{a}^{i_{1:m-1}} \sim \bar{\pi}^{i_{1:m-1}}}\left[\mathbb{E}_{a^{i_m} \sim \pi^{i_m}}\left[A_{\pi}^{i_m}\left(s, \mathbf{a}^{i_{1:m-1}}, a^{i_m}\right)\right]\right] = 0. \quad (6)$$

Building on Lemma 1 and Definition 2, we can finally generalise Theorem 1 of TRPO to MARL.

**Lemma 2.** *Let $\pi$ be a joint policy. Then, for any joint policy $\bar{\pi}$, we have*

$$J(\bar{\pi}) \geq J(\pi) + \sum_{m=1}^{n}\left[L_{\pi}^{i_{1:m}}\left(\bar{\pi}^{i_{1:m-1}}, \bar{\pi}^{i_m}\right) - C D_{KL}^{max}(\pi^{i_m}, \bar{\pi}^{i_m})\right].$$

For proof see Appendix C.2. This lemma provides an idea about how a joint policy can be improved. Namely, by Equation (6), we know that if any agents were to set the values of the above summands $L_{\pi}^{i_{1:m}}(\bar{\pi}^{i_{1:m-1}}, \bar{\pi}^{i_m}) - C D_{KL}^{max}(\pi^{i_m}, \bar{\pi}^{i_m})$ by sequentially updating their policies, each of them can always make its summand be zero by making no policy update (i.e., $\bar{\pi}^{i_m} = \pi^{i_m}$). This implies that any positive update will lead to an increment in summation. Moreover, as there are $n$ agents making policy updates, the compound increment can be large, leading to a substantial improvement. Lastly, note that this property holds with no requirement on the specific order by which agents make their updates; this allows for flexible scheduling on the update order at each iteration. To summarise, we propose the following Algorithm 1. We want to highlight that the algorithm is

---

**Algorithm 1** Multi-Agent Policy Iteration with Monotonic Improvement Guarantee

---

1: Initialise the joint policy $\pi_0 = (\pi_0^1, \ldots, \pi_0^n)$.
2: **for** $k = 0, 1, \ldots$ **do**
3:     Compute the advantage function $A_{\pi_k}(s, \mathbf{a})$ for all state-(joint)action pairs $(s, \mathbf{a})$.
4:     Compute $\epsilon = \max_{s, \mathbf{a}}|A_{\pi_k}(s, \mathbf{a})|$ and $C = \frac{4\gamma\epsilon}{(1-\gamma)^2}$.
5:     Draw a permutaion $i_{1:n}$ of agents at random.
6:     **for** $m = 1 : n$ **do**
7:         Make an update $\pi_{k+1}^{i_m} = \arg\max_{\pi^{i_m}}\left[L_{\pi_k}^{i_{1:m}}\left(\pi_{k+1}^{i_{1:m-1}}, \pi^{i_m}\right) - C D_{KL}^{max}(\pi_k^{i_m}, \pi^{i_m})\right]$.
8:     **end for**
9: **end for**

---

markedly different from naively applying the TRPO update, *i.e.*, Equation (3), on the joint policy of all agents. Firstly, our Algorithm 1 does not update the entire joint policy at once, but rather update each agent's individual policy sequentially. Secondly, during the sequential update, each agent has a unique optimisation objective that takes into account all previous agents' updates, which is also the key for the monotonic improvement property to hold.

## 3.2 THEORETICAL ANALYSIS

Now we justify by the following theorm that Algorithm 1 enjoys monotonic improvement property.

**Theorem 2.** *A sequence $(\boldsymbol{\pi}_k)_{k=0}^\infty$ of joint policies updated by Algorithm 1 has the monotonic improvement property, i.e., $J(\boldsymbol{\pi}_{k+1}) \geq J(\boldsymbol{\pi}_k)$ for all $k \in \mathbb{N}$.*

For proof see Appendix C.2. With the above theorem, we finally claim a successful introduction of trust region learning to MARL, as this generalises the monotonic improvement property of TRPO. Moreover, we take a step further to study the convergence property of Algorithm 1. Before stating the result, we introduce the following solution concept.

**Definition 3.** *In a fully-cooperative game, a joint policy $\boldsymbol{\pi}_* = (\pi_*^1, \dots, \pi_*^n)$ is a Nash equilibrium (NE) if for every $i \in \mathcal{N}$, $\pi^i \in \Pi^i$ implies $J(\boldsymbol{\pi}_*) \geq J(\pi^i, \boldsymbol{\pi}_*^{-i})$.*

NE (Nash, 1951) is a well-established game-theoretic solution concept. Definition 3 characterises the equilibrium point at convergence for cooperative MARL tasks. Based on this, we have the following result that describes Algorithm 1's asymptotic convergence behaviour towards NE.

**Theorem 3.** *Supposing in Algorithm 1 any permutation of agents has a fixed non-zero probability to begin the update, a sequence $(\boldsymbol{\pi}_k)_{k=0}^\infty$ of joint policies generated by the algorithm, in a cooperative Markov game, has a non-empty set of limit points, each of which is a Nash equilibrium.*

For proof see Appendix C.3. In deriving this result, the novel details introduced by Algorithm 1 played an important role. The monotonic improvement property (Theorem 2), achieved through the multi-agent advantage and the sequential update scheme, provided us with a guarantee on the convergence of the return. Furthermore, randomisation of the update order assured that, at convergence, none of the agents is incentified to make an update. The proof is finalised by excluding a possibility that the algorithm converges at non-equilibrium points.

## 4 PRACTICAL ALGORITHMS

When implementing Algorithm 1 in practice, large state and action spaces could prevent agents from designating policies $\pi^i(\cdot|s)$ for each state $s$ separately. To handle this, we parameterise each agent's policy $\pi_{\theta^i}^i$ by $\theta^i$, which, together with other agents' policies, forms a joint policy $\boldsymbol{\pi}_{\boldsymbol{\theta}}$ parametrised by $\boldsymbol{\theta} = (\theta^1, \dots, \theta^n)$. In this section, we develop two deep MARL algorithms to optimise the $\boldsymbol{\theta}$.

### 4.1 HATRPO

Computing $\mathrm{D}_{\mathrm{KL}}^{\max}(\pi_{\theta_k^{i_m}}^{i_m}, \pi_{\theta^{i_m}}^{i_m})$ in Algorithm 1 is challenging; it requires evaluating the KL-divergence for all states at each iteration. Similar to TRPO, one can ease this maximal KL-divergence penalty $\mathrm{D}_{\mathrm{KL}}^{\max}(\pi_{\theta_k^{i_m}}^{i_m}, \pi_{\theta^{i_m}}^{i_m})$ by replacing it with the expected KL-divergence constraint $\mathbb{E}_{s \sim \rho_{\boldsymbol{\pi}_{\boldsymbol{\theta}_k}}}\left[\mathrm{D}_{\mathrm{KL}}(\pi_{\theta_k^{i_m}}^{i_m}(\cdot|s), \pi_{\theta^{i_m}}^{i_m}(\cdot|s))\right] \leq \delta$ where $\delta$ is a threshold hyperparameter, and the expectation can be easily approximated by stochastic sampling. With the above amendment, we propose practical HATRPO algorithm in which, at every iteration $k+1$, given a permutation of agents $i_{1:n}$, agent $i_{m \in \{1, \dots, n\}}$ sequentially optimises its policy parameter $\theta_{k+1}^{i_m}$ by maximising a constrained objective:

$$\theta_{k+1}^{i_m} = \arg\max_{\theta^{i_m}} \mathbb{E}_{s \sim \rho_{\boldsymbol{\pi}_{\boldsymbol{\theta}_k}}, \mathbf{a}^{i_{1:m-1}} \sim \boldsymbol{\pi}_{\theta_{k+1}^{i_{1:m-1}}}^{i_{1:m-1}}, a^{i_m} \sim \pi_{\theta^{i_m}}^{i_m}}\left[A_{\boldsymbol{\pi}_{\boldsymbol{\theta}_k}}^{i_m}(s, \mathbf{a}^{i_{1:m-1}}, a^{i_m})\right],$$

$$\text{subject to } \mathbb{E}_{s \sim \rho_{\boldsymbol{\pi}_{\boldsymbol{\theta}_k}}}\left[\mathrm{D}_{\mathrm{KL}}(\pi_{\theta_k^{i_m}}^{i_m}(\cdot|s), \pi_{\theta^{i_m}}^{i_m}(\cdot|s))\right] \leq \delta. \tag{7}$$

To compute the above equation, similar to TRPO, one can apply a linear approximation to the objective function and a quadratic approximation to the KL constraint; the optimisation problem in Equation (7) can be solved by a closed-form update rule as

$$\theta_{k+1}^{i_m} = \theta_k^{i_m} + \alpha^j \sqrt{\frac{2\delta}{\boldsymbol{g}_k^{i_m}(\boldsymbol{H}_k^{i_m})^{-1}\boldsymbol{g}_k^{i_m}}}(\boldsymbol{H}_k^{i_m})^{-1}\boldsymbol{g}_k^{i_m}, \tag{8}$$

where $\boldsymbol{H}_k^{i_m} = \nabla^2_{\theta^{i_m}}\mathbb{E}_{s \sim \rho_{\boldsymbol{\pi}_{\boldsymbol{\theta}_k}}}\left[\mathrm{D}_{\mathrm{KL}}(\pi_{\theta_k^{i_m}}^{i_m}(\cdot|s), \pi_{\theta^{i_m}}^{i_m}(\cdot|s))\right]\big|_{\theta^{i_m} = \theta_k^{i_m}}$ is the Hessian of the expected KL-divergence, $\boldsymbol{g}_k^{i_m}$ is the gradient of the objective in Equation (7), $\alpha^j < 1$ is a positive coefficient that is found via backtracking line search, and the product of $(\boldsymbol{H}_k^{i_m})^{-1}\boldsymbol{g}_k^{i_m}$ can be efficiently computed with conjugate gradient algorithm.

The last missing piece for HATRPO is to estimate $\mathbb{E}_{\mathbf{a}^{i_{1:m-1}} \sim \boldsymbol{\pi}_{\theta_{k+1}^{i_{1:m-1}}}^{i_{1:m-1}}, a^{i_m} \sim \pi_{\theta^{i_m}}^{i_m}}\left[A_{\boldsymbol{\pi}_{\boldsymbol{\theta}_k}}^{i_m}(s, \mathbf{a}^{i_{1:m-1}}, a^{i_m})\right]$, which poses new challenges because each agent's objective has to take into account all previous

agents' updates, and the size of input vaires. Fortunately, with the following proposition, we can efficiently estimate this objective by employing a joint advantage estimator.

**Proposition 2.** *Let $\boldsymbol{\pi} = \prod_{j=1}^{n} \pi^{i_j}$ be a joint policy, and $A_{\boldsymbol{\pi}}(s, \mathbf{a})$ be its joint advantage function. Let $\bar{\boldsymbol{\pi}}^{i_{1:m-1}} = \prod_{j=1}^{m-1} \bar{\pi}^{i_j}$ be some **other** joint policy of agents $i_{1:m-1}$, and $\hat{\pi}^{i_m}$ be some **other** policy of agent $i_m$. Then, for every state $s$,*

$$\mathbb{E}_{\mathbf{a}^{i_{1:m-1}} \sim \bar{\boldsymbol{\pi}}^{i_{1:m-1}}, \mathbf{a}^{i_m} \sim \hat{\pi}^{i_m}} \left[ A_{\boldsymbol{\pi}}^{i_m} \left( s, \mathbf{a}^{i_{1:m-1}}, \mathbf{a}^{i_m} \right) \right]$$
$$= \mathbb{E}_{\mathbf{a} \sim \boldsymbol{\pi}} \left[ \left( \frac{\hat{\pi}^{i_m}(\mathbf{a}^{i_m}|s)}{\pi^{i_m}(\mathbf{a}^{i_m}|s)} - 1 \right) \frac{\bar{\boldsymbol{\pi}}^{i_{1:m-1}}(\mathbf{a}^{i_{1:m-1}}|s)}{\boldsymbol{\pi}^{i_{1:m-1}}(\mathbf{a}^{i_{1:m-1}}|s)} A_{\boldsymbol{\pi}}(s, \mathbf{a}) \right]. \quad (9)$$

For proof see Appendix D.1. One benefit of applying Equation (9) is that agents only need to maintain a joint advantage estimator $A_{\boldsymbol{\pi}}(s, \mathbf{a})$ rather than one centralised critic for each individual agent (e.g., unlike CTDE methods such as MADDPG). Another practical benefit one can draw is that, given an estimator $\hat{A}(s, a)$ of the advantage function $A_{\boldsymbol{\pi}_{\boldsymbol{\theta}_k}}(s, a)$, for example GAE (Schulman et al., 2015b), we can estimate $\mathbb{E}_{\mathbf{a}^{i_{1:m-1}} \sim \boldsymbol{\pi}_{\boldsymbol{\theta}_{k+1}}^{i_{1:m-1}}, \mathbf{a}^{i_m} \sim \pi_{\theta^{i_m}}^{i_m}} \left[ A_{\boldsymbol{\pi}_{\boldsymbol{\theta}_k}}^{i_m} \left( s, \mathbf{a}^{i_{1:m-1}}, \mathbf{a}^{i_m} \right) \right]$ with an estimator of

$$\left( \frac{\pi_{\boldsymbol{\theta}}^{i_m}(\mathbf{a}^{i_m}|s)}{\pi_{\boldsymbol{\theta}_k}^{i_m}(\mathbf{a}^{i_m}|s)} - 1 \right) M^{i_{1:m}}(s, \mathbf{a}), \quad \text{where } M^{i_{1:m}} = \frac{\bar{\boldsymbol{\pi}}^{i_{1:m-1}}(\mathbf{a}^{i_{1:m-1}}|s)}{\boldsymbol{\pi}^{i_{1:m-1}}(\mathbf{a}^{i_{1:m-1}}|s)} \hat{A}(s, \mathbf{a}). \quad (10)$$

Notably, Equation (10) aligns nicely with the sequential update scheme in HATRPO. For agent $i_m$, since previous agents $i_{1:m-1}$ have already made their updates, the compound policy ratio for $M^{i_{1:m}}$ in Equation (10) is easy to compute. Given a batch $\mathcal{B}$ of trajectories with length $T$, we can estimate the gradient with respect to policy parameters (derived in Appendix D.2) as follows,

$$\hat{\boldsymbol{g}}_k^{i_m} = \frac{1}{|\mathcal{B}|} \sum_{\tau \in \mathcal{B}} \sum_{t=0}^{T} M^{i_{1:m}}(s_t, \mathbf{a}_t) \nabla_{\theta^{i_m}} \log \pi_{\theta^{i_m}}^{i_m}(\mathbf{a}_t^i|s_t)\big|_{\theta^{i_m} = \theta_k^{i_m}}.$$

The term $-1 \cdot M^{i_{1:m}}(s, \mathbf{a})$ of Equation (10) is not reflected in $\hat{\boldsymbol{g}}_k^{i_m}$, as it only introduces a constant with zero gradient. Along with the Hessian of the expected KL-divergence, *i.e.*, $\boldsymbol{H}_k^{i_m}$, we can update $\theta_{k+1}^{i_m}$ by following Equation (8). The detailed pseudocode of HATRPO is listed in Appendix D.3.

## 4.2 HAPPO

To further alleviate the computation burden from $\boldsymbol{H}_k^{i_m}$ in HATRPO, one can follow the idea of PPO in Equation (4) by considering only using first order derivatives. This is achieved by making agent $i_m$ choose a policy parameter $\theta_{k+1}^{i_m}$ which maximises the clipping objective of

$$\mathbb{E}_{s \sim \rho_{\boldsymbol{\pi}_{\boldsymbol{\theta}_k}}, \mathbf{a} \sim \boldsymbol{\pi}_{\boldsymbol{\theta}_k}} \left[ \min \left( \frac{\pi_{\theta^{i_m}}^{i_m}(\mathbf{a}^i|s)}{\pi_{\theta_k^{i_m}}^{i_m}(\mathbf{a}^i|s)} M^{i_{1:m}}(s, \mathbf{a}), \text{clip}\left( \frac{\pi_{\theta^{i_m}}^{i_m}(\mathbf{a}^i|s)}{\pi_{\theta_k^{i_m}}^{i_m}(\mathbf{a}^i|s)}, 1 \pm \epsilon \right) M^{i_{1:m}}(s, \mathbf{a}) \right) \right]. \quad (11)$$

The optimisation process can be performed by stochastic gradient methods such as Adam (Kingma & Ba, 2014). We refer to the above procedure as HAPPO and Appendix D.4 for its full pseudocode.

## 4.3 RELATED WORK

We are fully aware of previous attempts that tried to extend TRPO/PPO into MARL. Despite empirical successes, none of them managed to propose a theoretically-justified trust region protocol in multi-agent learning, or maintain the monotonic improvement property. Instead, they tend to impose certain assumptions to enable direct implementations of TRPO/PPO in MARL problems. For example, IPPO (de Witt et al., 2020a) assume homogeneity of action spaces for all agents and enforce parameter sharing. Yu et al. (2021) proposed MAPPO which enhances IPPO by considering a joint critic function and finer implementation techniques for on-policy methods. Yet, it still suffers similar drawbacks of IPPO due to the lack of monotonic improvement guarantee especially when the parameter-sharing condition is switched off. Wen et al. (2021) adjusted PPO for MARL by considering a game-theoretical approach at the meta-game level among agents. Unfortunately, it can only deal with two-agent cases due to the intractability of Nash equilibrium. Recently, Li & He (2020) tried to implement TRPO for MARL through distributed consensus optimisation; however, they enforced the same ratio $\bar{\pi}^i(a^i|s)/\pi^i(a^i|s)$ for all agents (see their Equation (7)), which, similar

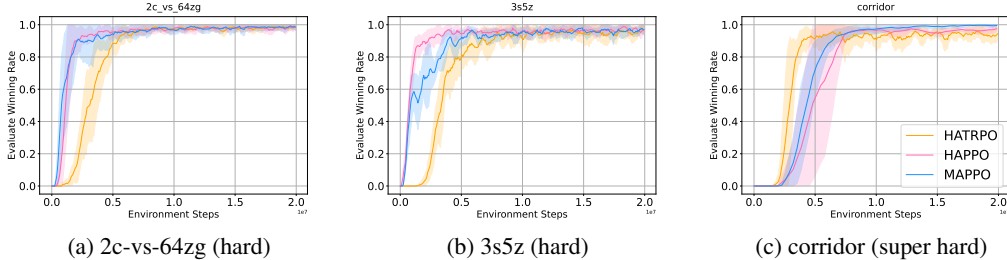

Figure 2: Performance comparisons between HATRPO/HAPPO and MAPPO on three SMAC tasks. Since all methods achieve 100% win rate, we believe SMAC is not sufficiently difficult to discriminate the capabilities of these algorithms, especially when non-parameter sharing is not required.

to parameter sharing, largely limits the policy space for optimisation. Moreover, their method comes with a $\delta/n$ KL-constraint threshold that fails to consider scenarios with large agent number.

One of the key ideas behind our HATRPO/HAPPO is the sequential update scheme. A similar idea of multi-agent sequential update was also discussed in the context of dynamic programming (Bertsekas, 2019) where artificial "in-between" states have to be considered. On the contrary, our sequential update sceheme is developed based on Lemma 1, which does not require any artificial assumptions and hold for any cooperative games. Furthermore, Bertsekas (2019) requires to maintain a fixed order of updates that is pre-defined for the task, whereas the order in HATRPO/MAPPO can be randomised at each iteration, which also offers desirable convergence property, as stated in Proposition 3 and also verified through ablation studies in Appendix F. The idea of sequential update also appeared in principal component analysis; in EigenGame (Gemp et al., 2020) eigenvectors, represented as players, maximise their own utility functions one-by-one. Although EigenGame provably solves the PCA problem, it is of little use in MARL, where a single iteration of sequential updates is insufficient to learn complex policies. Furthermore, its design and analysis rely on closed-form matrix calculus, which has no extension to MARL.

Lastly, we would like to highlight the importance of the decomposition result in Lemma 1. This result could serve as an effective solution to value-based methods in MARL where tremendous efforts have been made to decompose the joint Q-function into individual Q-functions when the joint Q-function are decomposable (Rashid et al., 2018). Lemma 1, in contrast, is a general result that holds for any cooperative MARL problems regardless of decomposibility. As such, we think of it as an appealing contribution to future developments on value-based MARL methods.

## 5 EXPERIMENTS AND RESULTS

We consider two most common benchmarks—StarCraftII Multi-Agent Challenge (SMAC) (Samvelyan et al., 2019) and Multi-Agent MuJoCo (de Witt et al., 2020b)—for evaluating MARL algorithms. All hyperparameter settings and implementations details can be found in Appendix E.

**StarCraftII Multi-Agent Challenge (SMAC)**. SMAC contains a set of StarCraft maps in which a team of ally units aims to defeat the opponent team. IPPO (de Witt et al., 2020a) and MAPPO (Yu et al., 2021) are known to achieve supreme results on this benchmark. By adopting parameter sharing, these methods achieve a winning rate of 100% on most maps, even including the maps that have heterogeneous agents. Therefore, we hypothesise that non-parameter sharing is **not necessarily** required and the trick of sharing policies is sufficient to solve SMAC tasks. We test our methods on two hard maps and one super-hard; results on Figure 2 confirm that SMAC is not sufficiently difficult to show off the capabilities of HATRPO/HAPPO when compared against existing methods.

**Multi-Agent MuJoCo**. In comparison to SMAC, we believe Mujoco enviornment provides a more suitable testing case for our methods. MuJoCo tasks challenge a robot to learn an optimal way of motion; Multi-Agent MuJoCo models each part of a robot as an independent agent, for example, a leg for a spider or an arm for a swimmer. With the increasing variety of the body parts, modelling heterogeneous policies becomes **necessary**. Figure 3 demonstrate that, in all scenarios, HATRPO and HAPPO enjoy superior performance over those of parameter-sharing methods: IPPO and MAPPO, and also outperform non-parameter sharing MADDPG (Lowe et al., 2017b) both in terms of reward values and variance. It is also worth noting that the performance gap between HATRPO and its rivals enlarges with the increasing number of agents. Meanwhile, we can observe that HATRPO outperforms HAPPO in almost all tasks; we believe it is because the hard KL constraint

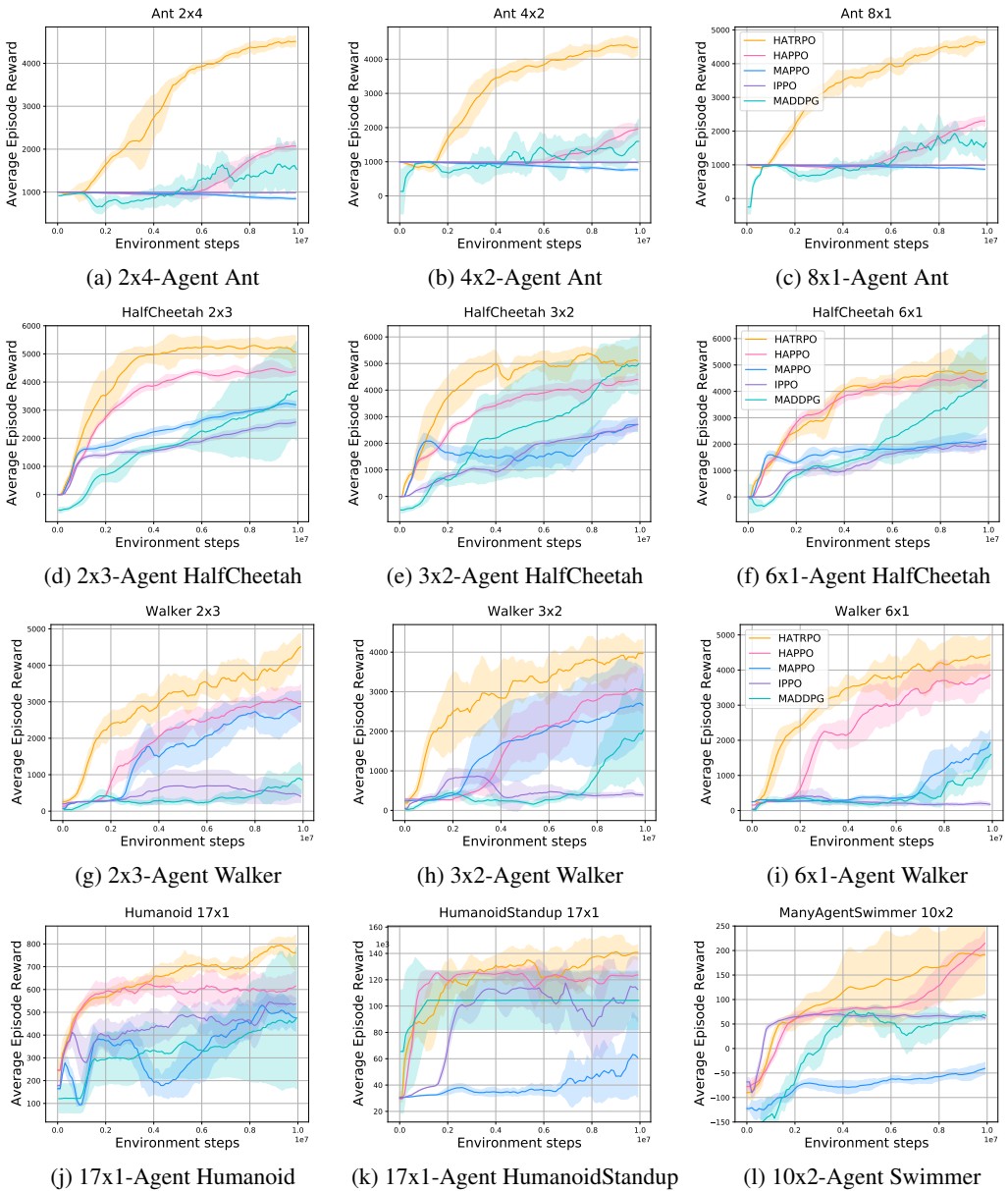

Figure 3: Performance comparison on multiple Multi-Agent MuJoCo tasks. HAPPO and HATRPO consistently outperform their rivals, thus establishing a new state-of-the-art algorithm for MARL. The performance gap enlarges with increasing number of agents.

in HATRPO, compared to the clipping version in HAPPO, relates more closely to Algorithm 1 that attains monotonic improvement guarantee.

## 6 CONCLUSION

In this paper, we successfully apply trust region learning to multi-agent settings by proposing the first MARL algorithm that attains theoretically-justified monotonical improvement property. The key to our development is the multi-agent advantage decomposition lemma that holds in general with no need for any assumptions on agents sharing parameters or the joint value function being decomposable. Based on this, we introduced two practical deep MARL algorithms: HATRPO and HAPPO. Experimental results on both discrete and continuous control tasks (*i.e.*, SMAC and Multi-Agent Mujoco) confirm their state-of-the-art performance. For future work, we will consider incorporating the safety constraint into HATRPO/HAPPO and propose rigorous safety-aware MARL solutions.

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

# Appendices

## A PRELIMINARIES

### A.1 DEFINITIONS AND ASSUMPTIONS

**Assumption 1.** *There exists $\eta \in \mathbb{R}$, such that $0 < \eta \ll 1$, and for every agent $i \in \mathcal{N}$, the policy space $\Pi^i$ is $\eta$-soft; that means that for every $\pi^i \in \Pi^i$, $s \in \mathcal{S}$, and $a^i \in \mathcal{A}^i$, we have $\pi^i(a^i|s) \geq \eta$.*

**Definition 3.** *In a fully-cooperative game, a joint policy $\boldsymbol{\pi}_* = (\pi_*^1, \ldots, \pi_*^n)$ is a Nash equilibrium (NE) if for every $i \in \mathcal{N}$, $\pi^i \in \Pi^i$ implies $J(\boldsymbol{\pi}_*) \geq J(\pi^i, \boldsymbol{\pi}_*^{-i})$.*

**Definition 4.** *Let $X$ be a finite set and $p : X \to \mathbb{R}$, $q : X \to \mathbb{R}$ be two maps. Then, the notion of **distance** between $p$ and $q$ that we adopt is given by $||p - q|| \triangleq \max_{x \in X} |p(x) - q(x)|$.*

### A.2 PROOFS OF PRELIMINARY RESULTS

**Lemma 3.** *Every agent $i$'s policy space $\Pi^i$ is convex and compact under the maximum norm.*

*Proof.* We start from proving convexity od the policy space: we prove that, for any two policies $\pi^i, \bar{\pi}^i \in \Pi^i$, for every $\alpha \in [0,1]$, $\alpha\pi^i + (1-\alpha)\bar{\pi}^i \in \Pi^i$. Clearly, for all $s \in \mathcal{S}$ and $a^i \in \mathcal{A}^i$, we have

$$\alpha\pi^i(a^i|s) + (1-\alpha)\bar{\pi}^i(a^i|s) \geq \alpha\eta + (1-\alpha)\eta = \eta.$$

Also, for every state $s$,

$$\sum_{a^i}\left[\alpha\pi^i(a^i|s) + (1-\alpha)\bar{\pi}^i(a^i|s)\right] = \alpha\sum_{a^i}\pi^i(a^i|s) + (1-\alpha)\sum_{a^i}\bar{\pi}^i(a^i|s) = \alpha + 1 - \alpha = 1,$$

which establishes convexity.

For compactness, we first prove that $\Pi^i$ is closed.

Let $\left(\pi_k^i\right)_{k=0}^{\infty}$ be a convergent sequence of policies of agent $i$. Let $\pi^i$ be the limit. We will prove that $\pi^i$ is a policy. First, by Assumption 1, for any $k \in \mathbb{N}$, $s \in \mathcal{S}$, and $a^i \in \mathcal{A}^i$, we have $\pi_k^i(a^i|s) \geq \eta$. Hence,

$$\pi^i(a^i|s) = \lim_{k \to \infty} \pi_k^i(a^i|s) \geq \lim_{k \to \infty} \eta \geq \eta.$$

Furtheremore, for any $k$ and $s$, we have $\sum_{a^i} \pi_k^i(a^i|s) = 1$. Hence

$$\sum_{a^i} \pi^i(a^i|s) = \sum_{a^i} \lim_{k \to \infty} \pi^i(a^i|s) = \lim_{k \to \infty} \sum_{a^i} \pi^i(a^i|s) = \lim_{k \to \infty} 1 = 1.$$

With these two conditions satisfied, $\pi^i$ is a policy, which proves the closure.

Further, for all policies $\pi^i$, states $s$ and actions $a$, $|\pi^i(a^i|s)| \leq 1$. This means that $||\pi^i||_{\max} \leq 1$, which proves boundedness. Hence, $\Pi^i$ is compact. $\square$

**Lemma 4** (Continuity of $\rho_\pi$). *The improper state distribution $\rho_\pi$ is continuous in $\pi$.*

*Proof.* First, let us show that for any $t \in \mathbb{N}$, the distribution $\rho_\pi^t$ is continous in $\pi$. We will do it by induction. This obviously holds when $t = 0$, as $\rho^0$ does not depend on policy. Hence, we can assume that for some $t \in \mathbb{N}$, the distribution $\rho_\pi^t$ is continuous in $\pi$. Let us now consider two policies $\pi$ and $\hat{\pi}$. Let $s' \in \mathcal{S}$. We have

$$\left|\rho_\pi^{t+1}(s') - \rho_{\hat{\pi}}^{t+1}(s')\right| = \left|\sum_s \rho_\pi^t(s)\sum_a \pi(a|s)P(s'|s,a) - \sum_s \rho_{\hat{\pi}}^t(s)\sum_a \hat{\pi}(a|s)P(s'|s,a)\right|$$

$$= \left|\sum_s\sum_a \left[\rho_\pi^t(s)\pi(a|s) - \rho_{\hat{\pi}}^t(s)\hat{\pi}(a|s)\right]P(s'|s,a)\right|$$

$$\leq \sum_s\sum_a \left|\rho_\pi^t(s)\pi(a|s) - \rho_{\hat{\pi}}^t(s)\hat{\pi}(a|s)\right|P(s'|s,a)$$

$$\begin{aligned}
&= \sum_s \sum_a \left| \rho_\pi^t(s)\pi(a|s) - \rho_\pi^t(s)\hat\pi(a|s) + \rho_\pi^t(s)\hat\pi(a|s) - \rho_{\hat\pi}^t(s)\hat\pi(a|s) \right| P(s'|s,a) \\
&\leq \sum_s \sum_a \left( \rho_\pi^t(s) \left| \pi(a|s) - \hat\pi(a|s) \right| + \hat\pi(a|s) \left| \rho_\pi^t(s) - \rho_{\hat\pi}^t(s) \right| \right) \\
&\leq \sum_s \sum_a \left( \rho_\pi^t(s)||\pi - \hat\pi|| + \hat\pi(a|s)||\rho_\pi^t - \rho_{\hat\pi}^t|| \right) \\
&= \sum_s \rho_\pi^t(s) \sum_a ||\pi - \hat\pi|| + \sum_s ||\rho_\pi^t - \rho_{\hat\pi}^t|| \sum_a \hat\pi(a|s) \\
&= |\mathcal{A}| \cdot ||\pi - \hat\pi|| + |\mathcal{S}| \cdot ||\rho_\pi^t - \rho_{\hat\pi}^t||.
\end{aligned}$$

Hence, we obtain

$$||\rho_\pi^{t+1} - \rho_{\hat\pi}^{t+1}|| \leq |\mathcal{A}| \cdot ||\pi - \bar\pi|| + |\mathcal{S}| \cdot ||\rho_\pi^t - \rho_{\hat\pi}^t||. \tag{12}$$

Using the base case, taking the limit as $\bar\pi \to \pi$, we get that the right-hand-side of Equation (12) converges to 0, which proves that every $\rho_\pi^{t+1}$ is continuous in $\pi$, and finishes the inductive step.

We can now prove that the total marginal state distribution $\rho_\pi$ is continous in $\pi$. To do that, let's take an arbitrary $\epsilon > 0$, and a natural $T$ such that $T > \frac{\log\left[\frac{\epsilon(1-\gamma)}{4}\right]}{\log\gamma}$. Equivalently, we choose $T$ such that $\frac{2\gamma^T}{1-\gamma} < \frac{\epsilon}{2}$. Let $\pi$ and $\hat\pi$ be two policies. We have

$$\begin{aligned}
|\rho_\pi(s) - \rho_{\hat\pi}(s)| &= \left| \sum_{t=0}^\infty \gamma^t \left( \rho_\pi^t(s) - \rho_{\hat\pi}^t(s) \right) \right| \\
&= \left| \sum_{t=0}^{T-1} \gamma^t \left( \rho_\pi^t(s) - \rho_{\hat\pi}^t(s) \right) + \gamma^T \sum_{t=T}^\infty \gamma^{t-T} \left( \rho_\pi^t(s) - \rho_{\hat\pi}^t(s) \right) \right| \\
&\leq \sum_{t=0}^{T-1} \gamma^t \left| \rho_\pi^t(s) - \rho_{\hat\pi}^t(s) \right| + \gamma^T \sum_{t=T}^\infty \gamma^{t-T} \left| \rho_\pi^t(s) - \rho_{\hat\pi}^t(s) \right| \\
&\leq \sum_{t=0}^{T-1} \gamma^t \left| \rho_\pi^t(s) - \rho_{\hat\pi}^t(s) \right| + \gamma^T \sum_{t=T}^\infty 2\gamma^{t-T} \\
&= \sum_{t=0}^{T-1} \gamma^t \left| \rho_\pi^t(s) - \rho_{\hat\pi}^t(s) \right| + \frac{2\gamma^T}{1-\gamma} < \sum_{t=0}^{T-1} \gamma^t \left| \rho_\pi^t(s) - \rho_{\hat\pi}^t(s) \right| + \frac{\epsilon}{2} \\
&\leq \sum_{t=0}^{T-1} \left| \rho_\pi^t(s) - \rho_{\hat\pi}^t(s) \right| + \frac{\epsilon}{2} \leq \sum_{t=0}^{T-1} ||\rho_\pi^t - \rho_{\hat\pi}^t|| + \frac{\epsilon}{2}. \tag{13}
\end{aligned}$$

Now, by continuity of each of $\rho_\pi^t$, for $t = 0, 1, \ldots, T-1$, we have that there exists a $\delta > 0$ such that $||\pi - \hat\pi|| < \delta$ implies $||\rho_\pi^t - \rho_{\hat\pi}^t|| < \frac{\epsilon}{2T}$. Taking such $\delta$, by Equation (13), we get $||\rho_\pi - \rho_{\hat\pi}|| \leq \epsilon$, which finishes the proof. $\square$

**Lemma 5** (Continuity of $Q_\pi$). *Let $\pi$ be a policy. Then $Q_\pi(s,a)$ is Lipschitz-continuous in $\pi$.*

*Proof.* Let $\pi$ and $\hat\pi$ be two policies. Then we have

$|Q_\pi(s,a) - Q_{\hat\pi}(s,a)|$

$$\begin{aligned}
&= \left| \left( r(s,a) + \gamma \sum_{s'} \sum_a P(s'|s,a)\pi(a|s)Q_\pi(s,a) \right) - \left( r(s,a) + \gamma \sum_{s'} \sum_a P(s'|s,a)\hat\pi(a|s)Q_{\hat\pi}(s,a) \right) \right| \\
&= \gamma \left| \sum_{s'} \sum_a P(s'|s,a) \left[ \pi(a|s)Q_\pi(s,a) - \hat\pi(a|s)Q_{\hat\pi}(s,a) \right] \right| \\
&\leq \gamma \sum_{s'} \sum_a P(s'|s,a) \left| \pi(a|s)Q_\pi(s,a) - \hat\pi(a|s)Q_{\hat\pi}(s,a) \right| \\
&= \gamma \sum_{s'} \sum_a P(s'|s,a) \left| \pi(a|s)Q_\pi(s,a) - \hat\pi(a|s)Q_\pi(s,a) + \hat\pi(a|s)Q_\pi(s,a) - \hat\pi(a|s)Q_{\hat\pi}(s,a) \right| \\
&\leq \gamma \sum_{s'} \sum_a P(s'|s,a) \left( |\pi(a|s)Q_\pi(s,a) - \hat\pi(a|s)Q_\pi(s,a)| + |\hat\pi(a|s)Q_\pi(s,a) - \hat\pi(a|s)Q_{\hat\pi}(s,a)| \right)
\end{aligned}$$

$$= \gamma \sum_{s'} \sum_{a} P(s'|s,a) |\pi(a|s) - \hat{\pi}(a|s)| \cdot |Q_\pi(s,a)|$$

$$+ \gamma \sum_{s'} \sum_{a} P(s'|s,a)\hat{\pi}(a|s) |Q_\pi(s,a) - Q_{\hat{\pi}}(s,a)|$$

$$\leq \gamma \sum_{s'} \sum_{a} P(s'|s,a)||\pi - \hat{\pi}|| \cdot Q_{\max}$$

$$+ \gamma \sum_{s'} \sum_{a} P(s'|s,a)\hat{\pi}(a|s)||Q_\pi - Q_{\hat{\pi}}||$$

$$= \gamma \, Q_{\max} \cdot |\mathcal{A}| \cdot ||\pi - \hat{\pi}|| + \gamma||Q_\pi - Q_{\hat{\pi}}||$$

Hence, we get

$$||Q_\pi - Q_{\hat{\pi}}|| \leq \gamma \, Q_{\max} \cdot |\mathcal{A}| \cdot ||\pi - \hat{\pi}|| + \gamma||Q_\pi - Q_{\hat{\pi}}||,$$

which implies

$$||Q_\pi - Q_{\hat{\pi}}|| \leq \frac{\gamma \, Q_{\max} \cdot |\mathcal{A}| \cdot ||\pi - \hat{\pi}||}{1 - \gamma}. \tag{14}$$

Equation (14) establishes Lipschitz-continuity with a constant $\frac{\gamma \, Q_{\max} \cdot |\mathcal{A}|}{1 - \gamma}$. $\qquad\square$

**Corollary 1.** *From Lemma 5 we obtain that the following functions are Lipschitz-continuous in $\pi$:*

$$\text{the state value function } V_\pi = \sum_a \pi(a|s)Q_\pi(s,a),$$

$$\text{the advantage function } A_\pi(s,a) = Q_\pi(s,a) - V_\pi(s),$$

$$\text{and the expected total reward } J(\pi) = \mathbb{E}_{s \sim \rho^0} [V_\pi(s)].$$

**Lemma 6.** *Let $\pi$ and $\hat{\pi}$ be policies. The quantity $\mathbb{E}_{s \sim \rho_\pi, a \sim \hat{\pi}} [A_\pi(s,a)]$ is continuous in $\pi$.*

*Proof.* We have

$$\mathbb{E}_{s \sim \rho_\pi, a \sim \hat{\pi}} [A_\pi(s,a)] = \sum_s \sum_a \rho_\pi(s)\hat{\pi}(a|s)A_\pi(s,a).$$

Continuity follows from continuity of each $\rho_\pi(s)$ (Lemma 4) and $A_\pi(s,a)$ (Corollary 1). $\qquad\square$

**Corollary 2** (Continuity in MARL). *All the results about continuity in $\pi$ extend to MARL. Policy $\pi$ can be replaced with joint policy $\boldsymbol{\pi}$; as $\boldsymbol{\pi}$ is Lipschitz-continuous in agent $i$'s policy $\pi^i$, the above continuity results extend to conitnuity in $\pi^i$. Thus, we will quote them in our proofs for MARL.*

## B   PROOF OF PROPOSITION 1

**Proposition 1.** *Let's consider a fully-cooperative game with an even number of agents $n$, one state, and the joint action space $\{0, 1\}^n$, where the reward is given by $r(\mathbf{0}^{n/2}, \mathbf{1}^{n/2}) = r(\mathbf{1}^{n/2}, \mathbf{0}^{n/2}) = 1$, and $r(\boldsymbol{a}^{1:n}) = 0$ for all other joint actions. Let $J^*$ be the optimal joint reward, and $J^*_{share}$ be the optimal joint reward under the shared policy constraint. Then*

$$\frac{J^*_{share}}{J^*} = \frac{2}{2^n}.$$

*Proof.* Clearly $J^* = 1$. An optimal joint policy in this case is, for example, the deterministic policy with joint action $(\mathbf{0}^{n/2}, \mathbf{1}^{n/2})$.

Now, let the shared policy be $(\theta, 1-\theta)$, where $\theta$ determines the probability that an agent takes action 0. Then, the expected reward is

$$J(\theta) = \Pr\left(\boldsymbol{a}^{1:n} = (\mathbf{0}^{n/2}, \mathbf{1}^{n/2})\right) \cdot 1 + \Pr\left(\boldsymbol{a}^{1:n} = (\mathbf{1}^{n/2}, \mathbf{0}^{n/2})\right) \cdot 1 = 2 \cdot \theta^{n/2}(1 - \theta)^{n/2}.$$

In order to maximise $J(\theta)$, we must maximise $\theta(1 - \theta)$, or equivalently, $\sqrt{\theta(1 - \theta)}$. By the artithmetic-geometric means inequality, we have

$$\sqrt{\theta(1 - \theta)} \leq \frac{\theta + (1 - \theta)}{2} = \frac{1}{2},$$

where the equality holds if and only if $\theta = 1 - \theta$, that is $\theta = \frac{1}{2}$. In such case we have

$$J_{\text{share}}^* = J\left(\frac{1}{2}\right) = 2 \cdot 2^{-n/2} \cdot 2^{-n/2} = \frac{2}{2^n},$$

which finishes the proof. □

## C  DERIVATION AND ANALYSIS OF ALGORITHM 1

### C.1  RECAP OF EXISTING RESULTS

**Lemma 7** (Performance Difference). *Let $\bar{\pi}$ and $\pi$ be two policies. Then, the following identity holds,*

$$J(\bar{\pi}) - J(\pi) = \mathbb{E}_{s \sim \rho_{\bar{\pi}}, a \sim \bar{\pi}} \left[ A_\pi(s, a) \right].$$

*Proof.* See Kakade & Langford (2002) (Lemma 6.1) or Schulman et al. (2015a) (Appendix A). □

**Theorem 1.** *(Schulman et al., 2015a, Theorem 1)  Let $\pi$ be the current policy and $\bar{\pi}$ be the next candidate policy. We define $L_\pi(\bar{\pi}) = J(\pi) + \mathbb{E}_{s \sim \rho_\pi, a \sim \bar{\pi}} \left[ A_\pi(s, a) \right], D_{KL}^{max}(\pi, \bar{\pi}) = \max_s D_{KL}\left(\pi(\cdot|s), \bar{\pi}(\cdot|s)\right)$. Then the inequality of*

$$J(\bar{\pi}) \geq L_\pi(\bar{\pi}) - C D_{KL}^{max}(\pi, \bar{\pi}) \tag{2}$$

*holds, where $C = \frac{4\gamma \max_{s,a} |A_\pi(s,a)|}{(1-\gamma)^2}$.*

*Proof.* See Schulman et al. (2015a) (Appendix A and Equation (9) of the paper). □

### C.2  ANALYSIS OF TRAINING OF ALGORITHM 1

**Lemma 1** (Multi-Agent Advantage Decomposition). *In any cooperative Markov games, given a joint policy $\boldsymbol{\pi}$, for any state $s$, and any agent subset $i_{1:m}$, the below equations holds.*

$$A_{\boldsymbol{\pi}}^{i_{1:m}}\left(s, \boldsymbol{a}^{i_{1:m}}\right) = \sum_{j=1}^{m} A_{\boldsymbol{\pi}}^{i_j}\left(s, \boldsymbol{a}^{i_{1:j-1}}, a^{i_j}\right).$$

*Proof.* By the definition of multi-agent advantage function,

$$
\begin{aligned}
A_{\boldsymbol{\pi_\theta}}^{i_{1:m}}(s, \boldsymbol{a}^{i_{1:m}}) &= Q_{\boldsymbol{\pi_\theta}}^{i_{1:m}}(s, \boldsymbol{a}^{i_{1:m}}) - V_{\boldsymbol{\pi_\theta}}(s) \\
&= \sum_{k=1}^{m} \left[ Q_{\boldsymbol{\pi_\theta}}^{i_{1:k}}(s, \boldsymbol{a}^{i_{1:k}}) - Q_{\boldsymbol{\pi_\theta}}^{i_{1:k-1}}(s, \boldsymbol{a}^{i_{1:k-1}}) \right] \\
&= \sum_{k=1}^{m} A_{\boldsymbol{\pi_\theta}}^{i_k}(s, \boldsymbol{a}^{i_{1:k-1}}, a^{i_k}),
\end{aligned}
$$

which finishes the proof.
Note that a similar finding has been shown in Kuba et al. (2021). □

**Lemma 8.** *Let $\boldsymbol{\pi} = \prod_{i=1}^{n} \pi^i$ and $\bar{\boldsymbol{\pi}} = \prod_{i=1}^{n} \bar{\pi}^i$ be joint policies. Then*

$$D_{KL}^{max}\left(\boldsymbol{\pi}, \bar{\boldsymbol{\pi}}\right) \leq \sum_{i=1}^{n} D_{KL}^{max}\left(\pi^i, \bar{\pi}^i\right)$$

*Proof.* For any state $s$, we have

$$\mathrm{D_{KL}}\left(\boldsymbol{\pi}(\cdot|s), \bar{\boldsymbol{\pi}}(\cdot|s)\right) = \mathbb{E}_{\mathbf{a}\sim\boldsymbol{\pi}}\left[\log\boldsymbol{\pi}(\mathbf{a}|s) - \log\bar{\boldsymbol{\pi}}(\mathbf{a}|s)\right]$$

$$= \mathbb{E}_{\mathbf{a}\sim\boldsymbol{\pi}}\left[\log\left(\prod_{i=1}^{n}\pi^i(\mathbf{a}^i|s)\right) - \log\left(\prod_{i=1}^{n}\bar{\pi}^i(\mathbf{a}^i|s)\right)\right]$$

$$= \mathbb{E}_{\mathbf{a}\sim\boldsymbol{\pi}}\left[\sum_{i=1}^{n}\log\pi^i(\mathbf{a}^i|s) - \sum_{i=1}^{n}\log\bar{\pi}^i(\mathbf{a}^i|s)\right]$$

$$= \sum_{i=1}^{n}\mathbb{E}_{\mathbf{a}^i\sim\pi^i, \mathbf{a}^{-i}\sim\boldsymbol{\pi}^{-i}}\left[\log\pi^i(\mathbf{a}^i|s) - \log\bar{\pi}^i(\mathbf{a}^i|s)\right] = \sum_{i=1}^{n}\mathrm{D_{KL}}\left(\pi^i(\cdot|s), \bar{\pi}^i(\cdot|s)\right).$$

Now, taking maximum over $s$ on both sides yields

$$\mathrm{D_{KL}^{max}}\left(\boldsymbol{\pi}, \bar{\boldsymbol{\pi}}\right) \leq \sum_{i=1}^{n}\mathrm{D_{KL}^{max}}\left(\pi^i, \bar{\pi}^i\right),$$

as required. $\qquad\square$

**Lemma 2.** *Let $\boldsymbol{\pi}$ be a joint policy. Then, for any joint policy $\bar{\boldsymbol{\pi}}$, we have*

$$J(\bar{\boldsymbol{\pi}}) \geq J(\boldsymbol{\pi}) + \sum_{m=1}^{n}\left[L_{\boldsymbol{\pi}}^{i_{1:m}}\left(\bar{\boldsymbol{\pi}}^{i_{1:m-1}}, \bar{\pi}^{i_m}\right) - CD_{KL}^{max}(\pi^{i_m}, \bar{\pi}^{i_m})\right].$$

*Proof.* By Theorem 1

$$J(\bar{\boldsymbol{\pi}}) \geq L_{\boldsymbol{\pi}}(\bar{\boldsymbol{\pi}}) - C\mathrm{D_{KL}^{max}}(\boldsymbol{\pi}, \bar{\boldsymbol{\pi}})$$
$$= J(\boldsymbol{\pi}) + \mathbb{E}_{\mathbf{s}\sim\rho_{\boldsymbol{\pi}}, \mathbf{a}\sim\bar{\boldsymbol{\pi}}}\left[A_{\boldsymbol{\pi}}(\mathbf{s}, \mathbf{a})\right] - C\mathrm{D_{KL}^{max}}(\boldsymbol{\pi}, \bar{\boldsymbol{\pi}})$$

which by Lemma 1 equals

$$= J(\boldsymbol{\pi}) + \mathbb{E}_{\mathbf{s}\sim\rho_{\boldsymbol{\pi}}, \mathbf{a}\sim\bar{\boldsymbol{\pi}}}\left[\sum_{m=1}^{n}A_{\boldsymbol{\pi}}^{i_m}\left(\mathbf{s}, \mathbf{a}^{i_{1:m-1}}, \mathbf{a}^{i_m}\right)\right] - C\mathrm{D_{KL}^{max}}(\boldsymbol{\pi}, \bar{\boldsymbol{\pi}})$$

and by Lemma 8 this is at least

$$\geq J(\boldsymbol{\pi}) + \mathbb{E}_{\mathbf{s}\sim\rho_{\boldsymbol{\pi}}, \mathbf{a}\sim\bar{\boldsymbol{\pi}}}\left[\sum_{m=1}^{n}A_{\boldsymbol{\pi}}^{i_m}\left(\mathbf{s}, \mathbf{a}^{i_{1:m-1}}, \mathbf{a}^{i_m}\right)\right] - \sum_{m=1}^{n}C\mathrm{D_{KL}^{max}}(\pi^{i_m}, \bar{\pi}^{i_m})$$

$$= J(\boldsymbol{\pi}) + \sum_{m=1}^{n}\mathbb{E}_{\mathbf{s}\sim\rho_{\boldsymbol{\pi}}, \mathbf{a}^{i_{1:m-1}}\sim\bar{\boldsymbol{\pi}}^{i_{1:m-1}}, \mathbf{a}^{i_m}\sim\bar{\pi}^{i_m}}\left[A_{\boldsymbol{\pi}}^{i_m}\left(\mathbf{s}, \mathbf{a}^{i_{1:m-1}}, \mathbf{a}^{i_m}\right)\right] - \sum_{m=1}^{n}C\mathrm{D_{KL}^{max}}(\pi^{i_m}, \bar{\pi}^{i_m})$$

$$= J(\boldsymbol{\pi}) + \sum_{m=1}^{n}\left(L_{\boldsymbol{\pi}}^{i_{1:m}}\left(\bar{\boldsymbol{\pi}}^{i_{1:m-1}}, \bar{\pi}^{i_m}\right) - C\mathrm{D_{KL}^{max}}(\pi^{i_m}, \bar{\pi}^{i_m})\right).$$

$\qquad\square$

**Theorem 2.** *A sequence $(\boldsymbol{\pi}_k)_{k=0}^{\infty}$ of joint policies updated by Algorithm 1 has the monotonic improvement property, i.e., $J(\boldsymbol{\pi}_{k+1}) \geq J(\boldsymbol{\pi}_k)$ for all $k \in \mathbb{N}$.*

*Proof.* Let $\boldsymbol{\pi}_0$ be any joint policy. For every $k \geq 0$, the joint policy $\boldsymbol{\pi}_{k+1}$ is obtained from $\boldsymbol{\pi}_k$ by Algorithm 1 update; for $m = 1, \ldots, n$,

$$\pi_{k+1}^{i_m} = \underset{\pi^{i_m}}{\arg\max}\left[L_{\boldsymbol{\pi}_k}^{i_{1:m}}\left(\boldsymbol{\pi}_{k+1}^{i_{1:m-1}}, \pi^{i_m}\right) - C\mathrm{D_{KL}^{max}}\left(\pi_k^{i_m}, \pi^{i_m}\right)\right].$$

By Theorem 1, we have

$$J(\boldsymbol{\pi}_{k+1}) \geq L_{\boldsymbol{\pi}_k}(\boldsymbol{\pi}_{k+1}) - C\mathrm{D}_{\mathrm{KL}}^{\max}(\boldsymbol{\pi}_k, \boldsymbol{\pi}_{k+1}),$$

which by Lemma 8 is lower-bounded by

$$\geq L_{\boldsymbol{\pi}_k}(\boldsymbol{\pi}_{k+1}) - \sum_{m=1}^{n} C\mathrm{D}_{\mathrm{KL}}^{\max}(\pi_k^{i_m}, \pi_{k+1}^{i_m})$$

$$= J(\boldsymbol{\pi}_k) + \sum_{m=1}^{n} \left( L_{\boldsymbol{\pi}_k}^{i_{1:m}}(\boldsymbol{\pi}_{k+1}^{i_{1:m-1}}, \pi_{k+1}^{i_m}) - C\mathrm{D}_{\mathrm{KL}}^{\max}(\pi_k^{i_m}, \pi_{k+1}^{i_m}) \right), \tag{15}$$

and as for every $m$, $\pi_{k+1}^{i_m}$ is the argmax, this is lower-bounded by

$$\geq J(\boldsymbol{\pi}_k) + \sum_{m=1}^{n} \left( L_{\boldsymbol{\pi}_k}^{i_{1:m}}(\boldsymbol{\pi}_{k+1}^{i_{1:m-1}}, \pi_k^{i_m}) - C\mathrm{D}_{\mathrm{KL}}^{\max}(\pi_k^{i_m}, \pi_k^{i_m}) \right),$$

which, as mentioned in Definition 2, equals

$$= J(\boldsymbol{\pi}_k) + \sum_{m=1}^{n} 0 = J(\boldsymbol{\pi}_k),$$

where the last inequality follows from Equation (6). This proves that Algorithm 1 achieves monotonic improvement. □

## C.3 Analysis of Convergence of Algorithm 1

**Theorem 3.** *Supposing in Algorithm 1 any permutation of agents has a fixed non-zero probability to begin the update, a sequence $(\boldsymbol{\pi}_k)_{k=0}^{\infty}$ of joint policies generated by the algorithm, in a cooperative Markov game, has a non-empty set of limit points, each of which is a Nash equilibrium.*

*Proof.* **Step 1 (convergence).** Firstly, it is clear that the sequence $(J(\boldsymbol{\pi}_k))_{k=0}^{\infty}$ converges as, by Theorem 2, it is non-decreasing and bounded above by $\frac{R_{\max}}{1-\gamma}$. Let us denote the limit by $\bar{J}$. For every $k$, we denote the tuple of agents, according to whose order the agents perform the sequential updates, by $i_{1:n}^k$, and we note that $\left(i_{1:n}^k\right)_{k \in \mathbb{N}}$ is a random process. Furthermore, we know that the sequence of policies $(\boldsymbol{\pi}_k)$ is bounded, so by Bolzano-Weierstrass Theorem, it has at least one convergent subsequence. Let $\bar{\boldsymbol{\pi}}$ be any limit point of the sequence (note that the set of limit points is a random set), and $\left(\boldsymbol{\pi}_{k_j}\right)_{j=0}^{\infty}$ be a subsequence converging to $\bar{\boldsymbol{\pi}}$ (which is a random subsequence as well). By continuity of $J$ in $\boldsymbol{\pi}$ (Corollary 1), we have

$$J(\bar{\boldsymbol{\pi}}) = J\left(\lim_{j \to \infty} \boldsymbol{\pi}_{k_j}\right) = \lim_{j \to \infty} J\left(\boldsymbol{\pi}_{k_j}\right) = \bar{J}. \tag{16}$$

For now, we introduce an auxiliary definition.

**Definition 5** (TR-Stationarity). *A joint policy $\bar{\boldsymbol{\pi}}$ is trust-region-stationary (TR-stationary) if, for every agent $i$,*

$$\bar{\pi}^i = \arg\max_{\pi^i} \left[ \mathbb{E}_{\mathrm{s} \sim \rho_{\bar{\boldsymbol{\pi}}}, \mathrm{a}^i \sim \pi^i} \left[ A_{\bar{\boldsymbol{\pi}}}^i(\mathrm{s}, \mathrm{a}^i) \right] - C_{\bar{\boldsymbol{\pi}}} \mathrm{D}_{KL}^{max}\left(\bar{\pi}^i, \pi^i\right) \right],$$

*where $C_{\bar{\boldsymbol{\pi}}} = \frac{4\gamma\epsilon}{(1-\gamma)^2}$, and $\epsilon = \max_{s,\boldsymbol{a}} |A_{\bar{\boldsymbol{\pi}}}(s, \boldsymbol{a})|$.*

We will now establish the TR-stationarity of any limit point joint policy $\bar{\boldsymbol{\pi}}$ (which, as stated above, is a random variable). Let $\mathbb{E}_{i_{1:n}^{0:\infty}}[\cdot]$ denote the expected value operator under the random process $(i_{1:n}^{0:\infty})$. Let also $\epsilon_k = \max_{s,\boldsymbol{a}} |A_{\boldsymbol{\pi}_k}(s, \boldsymbol{a})|$, and $C_k = \frac{4\gamma\epsilon_k}{(1-\gamma)^2}$. We have

$$0 = \lim_{k \to \infty} \mathbb{E}_{i_{1:n}^{0:\infty}}\left[ J(\boldsymbol{\pi}_{k+1}) - J(\boldsymbol{\pi}_k) \right]$$

$$\geq \lim_{k \to \infty} \mathbb{E}_{i_{1:n}^{0:\infty}}\left[ L_{\boldsymbol{\pi}_k}(\boldsymbol{\pi}_{k+1}) - C_k \mathrm{D}_{\mathrm{KL}}^{\max}(\boldsymbol{\pi}_k, \boldsymbol{\pi}_{k+1}) \right] \text{ by Theorem 1}$$

$$\geq \lim_{k \to \infty} \mathbb{E}_{i_{1:n}^{0:\infty}}\left[ L_{\boldsymbol{\pi}_k}^{i_1^k}\left(\pi_{k+1}^{i_1^k}\right) - C_k \mathrm{D}_{\mathrm{KL}}^{\max}\left(\pi_k^{i_1^k}, \pi_{k+1}^{i_1^k}\right) \right]$$

by Equation (15) and the fact that each of its summands is non-negative

Now, we consider an arbitrary limit point $\bar{\boldsymbol{\pi}}$ from the (random) limit set, and a (random) subsequence $\left(\boldsymbol{\pi}_{k_j}\right)_{j=0}^{\infty}$ that converges to $\bar{\boldsymbol{\pi}}$. We get

$$0 \geq \lim_{j \to \infty} \mathbb{E}_{i_{1:n}^{0:\infty}} \left[ L_{\boldsymbol{\pi}_{k_j}}^{i_1^{k_j}} \left( \pi_{k_j+1}^{i_1^{k_j}} \right) - C_{k_j} \mathrm{D}_{\mathrm{KL}}^{\max} \left( \pi_{k_j}^{i_1^{k_j}}, \pi_{k_j+1}^{i_1^{k_j}} \right) \right].$$

As the expectation is taken of non-negative random variables, and for every $i \in \mathcal{N}$ and $k \in \mathbb{N}$, with some positive probability $p_i$, we have $i_1^{k_j} = i$ (because every permutation has non-zero probability), the above is bounded from below by

$$p_i \lim_{j \to \infty} \max_{\pi^i} \left[ L_{\boldsymbol{\pi}_{k_j}}^i (\pi^i) - C_{k_j} \mathrm{D}_{\mathrm{KL}}^{\max} \left( \pi_{k_j}^i, \pi^i \right) \right],$$

which, as $\boldsymbol{\pi}_{k_j}$ converges to $\bar{\boldsymbol{\pi}}$, equals to

$$p_i \max_{\pi^i} \left[ L_{\bar{\boldsymbol{\pi}}}^i (\pi^i) - C_{\bar{\boldsymbol{\pi}}} \mathrm{D}_{\mathrm{KL}}^{\max} \left( \bar{\pi}^i, \pi^i \right) \right] \geq 0, \quad \text{by Equation (6)}.$$

For convergence of $L_{\boldsymbol{\pi}_{k_j}}^i (\pi^i)$ we used Definition 2 combined with Lemma 6, for convergence of $C_{k_j}$ we used Corollary 1, and the convergence of $\mathrm{D}_{\mathrm{KL}}^{\max}$ follows from continuity of $\mathrm{D}_{\mathrm{KL}}$ and $\max$. This proves that, for any limit point $\bar{\boldsymbol{\pi}}$ of the random process $(\boldsymbol{\pi}_k)$ induced by Algorithm 1, $\max_{\pi^i} \left[ L_{\bar{\boldsymbol{\pi}}}^i (\pi^i) - C_{\bar{\boldsymbol{\pi}}} \mathrm{D}_{\mathrm{KL}}^{\max} \left( \bar{\pi}^i, \pi^i \right) \right] = 0$, which is equivalent with Definition 5.

**Step 2 (dropping the penalty term).** Now, we have to prove that TR-stationary points are NEs of cooperative Markov games. The main step is to prove the following statement: *a TR-stationary joint policy* $\bar{\boldsymbol{\pi}}$, *for every state* $s \in \mathcal{S}$, *satisfies*

$$\bar{\pi}^i = \arg \max_{\pi^i} \mathbb{E}_{\mathbf{a}^i \sim \pi^i} \left[ A_{\bar{\boldsymbol{\pi}}}^i (s, \mathbf{a}^i) \right]. \tag{17}$$

We will use the technique of the proof by contradiction. Suppose that there is a state $s_0$ such that there exists a policy $\hat{\pi}^i$ with

$$\mathbb{E}_{\mathbf{a}^i \sim \hat{\pi}^i} \left[ A_{\bar{\boldsymbol{\pi}}}^i (s_0, \mathbf{a}^i) \right] > \mathbb{E}_{\mathbf{a}^i \sim \bar{\pi}^i} \left[ A_{\bar{\boldsymbol{\pi}}}^i (s_0, \mathbf{a}^i) \right]. \tag{18}$$

Let us parametrise the policies $\pi^i$ according to the template

$$\pi^i(\cdot|s_0) = \left( x_1^i, \ldots, x_{d^i-1}^i, 1 - \sum_{j=1}^{d^i-1} x_j^i \right)$$

where the values of $x_j^i$ ($j = 1, \ldots, d^i - 1$) are such that $\pi^i(\cdot|s_0)$ is a valid probability distribution. Then we can rewrite our quantity of interest (the objective of Equation (17) as

$$\mathbb{E}_{\mathbf{a}^i \sim \pi^i} \left[ A_{\bar{\boldsymbol{\pi}}}^i (s_0, \mathbf{a}^i) \right] = \sum_{j=1}^{d^i-1} x_j^i \cdot A_{\bar{\pi}}^i \left( s_0, a_j^i \right) + \left( 1 - \sum_{h=1}^{d^i-1} x_h^i \right) A_{\bar{\boldsymbol{\pi}}}^i \left( s_0, a_{d^i}^i \right)$$

$$= \sum_{j=1}^{d^i-1} x_j^i \left[ A_{\bar{\pi}}^i \left( s_0, a_j^i \right) - A_{\bar{\boldsymbol{\pi}}}^i \left( s_0, a_{d^i}^i \right) \right] + A_{\bar{\boldsymbol{\pi}}}^i \left( s_0, a_{d^i}^i \right),$$

which is an affine function of the policy parameterisation. It follows that its gradient (with respect to $x^i$) and directional derivatives are constant in the space of policies at state $s_0$. The existance of policy $\hat{\pi}^i(\cdot|s_0)$, for which Inequality (18) holds, implies that the directional derivative in the direction from $\bar{\pi}^i(\cdot|s_0)$ to $\hat{\pi}^i(\cdot|s_0)$ is strictly positive. We also have

$$\frac{\partial \mathrm{D}_{\mathrm{KL}}(\bar{\pi}^i(\cdot|s_0), \pi^i(\cdot|s_0))}{\partial x_j^i} = \frac{\partial}{\partial x_j^i} \left[ (\bar{\pi}^i(\cdot|s_0))^T (\log \bar{\pi}^i(\cdot|s_0) - \log \pi^i(\cdot|s_0)) \right]$$

$$= \frac{\partial}{\partial x_j^i} \left[ -(\bar{\pi}^i)^T \log \pi^i \right] \text{ (omitting state } s_0 \text{ for brevity)}$$

$$= -\frac{\partial}{\partial x_j^i} \sum_{k=1}^{d_i-1} \bar{\pi}_k^i \log x_k^i - \frac{\partial}{\partial x_j^i} \bar{\pi}_{d_i}^i \log \left( 1 - \sum_{k=1}^{d_i-1} x_k^i \right)$$

$$= -\frac{\bar{\pi}_j^i}{x_j^i} + \frac{\bar{\pi}_{d_i}^i}{1 - \sum_{k=1}^{d_i-1} x_k^i}$$

$$= -\frac{\bar{\pi}_j^i}{\pi_j^i} + \frac{\bar{\pi}_{d_i}^i}{\pi_{d_i}^i} = 0, \quad \text{when evaluated at } \pi^i = \bar{\pi}^i, \tag{19}$$

which means that the KL-penalty has zero gradient at $\bar{\pi}^i(\cdot|s_0)$. Hence, when evaluated at $\pi^i(\cdot|s_0) = \bar{\pi}^i(\cdot|s_0)$, the objective

$$\rho_{\bar{\boldsymbol{\pi}}}(s_0)\mathbb{E}_{\mathrm{a}^i\sim\pi^i}\big[A^i_{\bar{\boldsymbol{\pi}}}(s_0,\mathrm{a}^i)\big] - C_{\bar{\boldsymbol{\pi}}}\mathrm{D}_{\mathrm{KL}}\big(\bar{\pi}^i(\cdot|s_0),\pi^i(\cdot|s_0)\big)$$

has a strictly positive directional derivative in the direction of $\hat{\pi}^i(\cdot|s_0)$. Thus, there exists a policy $\widetilde{\pi}^i(\cdot|s_0)$, sufficiently close to $\bar{\pi}^i(\cdot|s_0)$ on the path joining it with $\hat{\pi}^i(\cdot|s_0)$, for which

$$\rho_{\bar{\boldsymbol{\pi}}}(s_0)\mathbb{E}_{\mathrm{a}^i\sim\widetilde{\pi}^i}\big[A^i_{\bar{\boldsymbol{\pi}}}(s_0,\mathrm{a}^i)\big] - C_{\bar{\boldsymbol{\pi}}}\mathrm{D}_{\mathrm{KL}}\big(\bar{\pi}^i(\cdot|s_0),\widetilde{\pi}^i(\cdot|s_0)\big) > 0.$$

Let $\pi^i_*$ be a policy such that $\pi^i_*(\cdot|s_0) = \widetilde{\pi}^i(\cdot|s_0)$, and $\pi^i_*(\cdot|s) = \bar{\pi}^i(\cdot|s)$ for states $s \neq s_0$. As for these states we have

$$\rho_{\bar{\boldsymbol{\pi}}}(s)\mathbb{E}_{\mathrm{a}^i\sim\pi^i_*}\big[A^i_{\bar{\boldsymbol{\pi}}}(s,\mathrm{a}^i)\big] = \rho_{\bar{\boldsymbol{\pi}}}(s)\mathbb{E}_{\mathrm{a}^i\sim\bar{\pi}^i}\big[A^i_{\bar{\boldsymbol{\pi}}}(s,\mathrm{a}^i)\big] = 0, \text{ and } \mathrm{D}_{\mathrm{KL}}(\bar{\pi}^i(\cdot|s),\pi^i_*(\cdot|s)) = 0,$$

it follows that

$$
\begin{aligned}
L^i_{\bar{\boldsymbol{\pi}}}(\pi^i_*) - C_{\bar{\boldsymbol{\pi}}}\mathrm{D}^{\max}_{\mathrm{KL}}(\bar{\pi}^i,\pi^i_*) &= \rho_{\bar{\boldsymbol{\pi}}}(s_0)\mathbb{E}_{\mathrm{a}^i\sim\widetilde{\pi}^i}\big[A^i_{\bar{\boldsymbol{\pi}}}(s_0,\mathrm{a}^i)\big] - C_{\bar{\boldsymbol{\pi}}}\mathrm{D}_{\mathrm{KL}}\big(\bar{\pi}^i(\cdot|s_0),\widetilde{\pi}^i(\cdot|s_0)\big) \\
&> 0 = L^i_{\bar{\boldsymbol{\pi}}}(\bar{\pi}^i) - C_{\bar{\boldsymbol{\pi}}}\mathrm{D}^{\max}_{\mathrm{KL}}(\bar{\pi}^i,\bar{\pi}^i),
\end{aligned}
$$

which is a contradiction with TR-stationarity of $\bar{\boldsymbol{\pi}}$. Hence, the claim of Equation (17) is proved. **Step 3 (optimality).** Now, for a fixed joint policy $\bar{\boldsymbol{\pi}}^{-i}$ of other agents, $\bar{\pi}^i$ satisfies

$$\bar{\pi}^i = \arg\max_{\pi^i}\mathbb{E}_{\mathrm{a}^i\sim\pi^i}\big[A^i_{\bar{\boldsymbol{\pi}}}(s,\mathrm{a}^i)\big] = \arg\max_{\pi^i}\mathbb{E}_{\mathrm{a}^i\sim\pi^i}\big[Q^i_{\bar{\boldsymbol{\pi}}}(s,\mathrm{a}^i)\big], \ \forall s \in \mathcal{S},$$

which is the Bellman optimality equation (Sutton & Barto, 2018). Hence, for a fixed joint policy $\bar{\boldsymbol{\pi}}^{-i}$, the policy $\bar{\pi}^i$ is optimal:

$$\bar{\pi}^i = \arg\max_{\pi^i} J(\pi^i, \bar{\boldsymbol{\pi}}^{-i}).$$

As agent $i$ was chosen arbitrarily, $\bar{\boldsymbol{\pi}}$ is a Nash equilibrium. $\qquad\square$

# D  HATRPO AND HAPPO

## D.1  PROOF OF PROPOSITION 2

**Proposition 2.** *Let $\boldsymbol{\pi} = \prod_{j=1}^{n} \pi^{i_j}$ be a joint policy, and $A_{\boldsymbol{\pi}}(\mathrm{s}, \mathrm{a})$ be its joint advantage function. Let $\bar{\boldsymbol{\pi}}^{i_{1:m-1}} = \prod_{j=1}^{m-1} \bar{\pi}^{i_j}$ be some **other** joint policy of agents $i_{1:m-1}$, and $\hat{\pi}^{i_m}$ be some **other** policy of agent $i_m$. Then, for every state $s$,*

$$\mathbb{E}_{\mathbf{a}^{i_{1:m-1}} \sim \bar{\boldsymbol{\pi}}^{i_{1:m-1}}, a^{i_m} \sim \hat{\pi}^{i_m}} \left[ A_{\boldsymbol{\pi}}^{i_m} \left( s, \mathbf{a}^{i_{1:m-1}}, a^{i_m} \right) \right]$$

$$= \mathbb{E}_{\mathbf{a} \sim \boldsymbol{\pi}} \left[ \left( \frac{\hat{\pi}^{i_m}(a^{i_m}|s)}{\pi^{i_m}(a^{i_m}|s)} - 1 \right) \frac{\bar{\boldsymbol{\pi}}^{i_{1:m-1}}(\mathbf{a}^{i_{1:m-1}}|s)}{\boldsymbol{\pi}^{i_{1:m-1}}(\mathbf{a}^{i_{1:m-1}}|s)} A_{\boldsymbol{\pi}}(s, \mathbf{a}) \right]. \quad (9)$$

*Proof.* We have

$$= \mathbb{E}_{\mathbf{a} \sim \boldsymbol{\pi}} \left[ \frac{\bar{\boldsymbol{\pi}}^{i_{1:m}}(\mathbf{a}^{i_{1:m}}|s)}{\boldsymbol{\pi}^{i_{1:m}}(\mathbf{a}^{i_{1:m}}|s)} A_{\boldsymbol{\pi}}(s, \mathbf{a}) - \frac{\bar{\boldsymbol{\pi}}^{i_{1:m-1}}(\mathbf{a}^{i_{1:m-1}}|s)}{\boldsymbol{\pi}^{i_{1:m-1}}(\mathbf{a}^{i_{1:m-1}}|s)} A_{\boldsymbol{\pi}}(s, \mathbf{a}) \right]$$

$$= \mathbb{E}_{\mathbf{a}^{i_{1:m}} \sim \boldsymbol{\pi}^{i_{1:m}}, \mathbf{a}^{-i_{1:m}} \sim \boldsymbol{\pi}^{-i_{1:m}}} \left[ \frac{\bar{\boldsymbol{\pi}}^{i_{1:m}}(\mathbf{a}^{i_{1:m}}|s)}{\boldsymbol{\pi}^{i_{1:m}}(\mathbf{a}^{i_{1:m}}|s)} A_{\boldsymbol{\pi}}(s, \mathbf{a}^{i_{1:m}}, \mathbf{a}^{-i_{1:m}}) \right]$$

$$- \mathbb{E}_{\mathbf{a}^{i_{1:m-1}} \sim \boldsymbol{\pi}^{i_{1:m-1}}, \mathbf{a}^{-i_{1:m-1}} \sim \boldsymbol{\pi}^{-i_{1:m-1}}} \left[ \frac{\bar{\boldsymbol{\pi}}^{i_{1:m-1}}(\mathbf{a}^{i_{1:m-1}}|s)}{\boldsymbol{\pi}^{i_{1:m-1}}(\mathbf{a}^{i_{1:m-1}}|s)} A_{\boldsymbol{\pi}}(s, \mathbf{a}^{i_{1:m-1}}, \mathbf{a}^{-i_{1:m-1}}) \right]$$

$$= \mathbb{E}_{\mathbf{a}^{i_{1:m}} \sim \bar{\boldsymbol{\pi}}^{i_{1:m}}, \mathbf{a}^{-i_{1:m}} \sim \boldsymbol{\pi}^{-i_{1:m}}} \left[ A_{\boldsymbol{\pi}}(s, \mathbf{a}^{i_{1:m}}, \mathbf{a}^{-i_{1:m}}) \right]$$

$$- \mathbb{E}_{\mathbf{a}^{i_{1:m-1}} \sim \bar{\boldsymbol{\pi}}^{i_{1:m-1}}, \mathbf{a}^{-i_{1:m-1}} \sim \boldsymbol{\pi}^{-i_{1:m-1}}} \left[ A_{\boldsymbol{\pi}}(s, \mathbf{a}^{i_{1:m-1}}, \mathbf{a}^{-i_{1:m-1}}) \right]$$

$$= \mathbb{E}_{\mathbf{a}^{i_{1:m}} \sim \bar{\boldsymbol{\pi}}^{i_{1:m}}} \left[ \mathbb{E}_{\mathbf{a}^{-i_{1:m}} \sim \boldsymbol{\pi}^{-i_{1:m}}} \left[ A_{\boldsymbol{\pi}}(s, \mathbf{a}^{i_{1:m}}, \mathbf{a}^{-i_{1:m}}) \right] \right]$$

$$- \mathbb{E}_{\mathbf{a}^{i_{1:m-1}} \sim \bar{\boldsymbol{\pi}}^{i_{1:m-1}}} \left[ \mathbb{E}_{\mathbf{a}^{-i_{1:m-1}} \sim \boldsymbol{\pi}^{-i_{1:m-1}}} \left[ A_{\boldsymbol{\pi}}(s, \mathbf{a}^{i_{1:m-1}}, \mathbf{a}^{-i_{1:m-1}}) \right] \right]$$

$$= \mathbb{E}_{\mathbf{a}^{i_{1:m}} \sim \bar{\boldsymbol{\pi}}^{i_{1:m}}} \left[ A_{\boldsymbol{\pi}}^{i_{1:m}}(s, \mathbf{a}^{i_{1:m}}) \right]$$

$$- \mathbb{E}_{\mathbf{a}^{i_{1:m-1}} \sim \bar{\boldsymbol{\pi}}^{i_{1:m-1}}} \left[ A_{\boldsymbol{\pi}}^{i_{1:m-1}}(s, \mathbf{a}^{i_{1:m-1}}) \right],$$

which, by Lemma 1, equals

$$= \mathbb{E}_{\mathbf{a}^{i_{1:m}} \sim \bar{\boldsymbol{\pi}}^{i_{1:m}}} \left[ A_{\boldsymbol{\pi}}^{i_{1:m}}(s, \mathbf{a}^{i_{1:m}}) - A_{\boldsymbol{\pi}}^{i_{1:m-1}}(s, \mathbf{a}^{i_{1:m-1}}) \right]$$

$$= \mathbb{E}_{\mathbf{a}^{i_{1:m}} \sim \bar{\boldsymbol{\pi}}^{i_{1:m}}} \left[ A_{\boldsymbol{\pi}}^{i_m}(s, \mathbf{a}^{i_{1:m-1}}, a^{i_m}) \right].$$

$\square$

## D.2  DERIVATION OF THE GRADIENT ESTIMATOR FOR HATRPO

$$\nabla_{\theta^{i_m}} \mathbb{E}_{\mathrm{s} \sim \rho_{\boldsymbol{\pi}_{\theta_k}}, \mathbf{a} \sim \boldsymbol{\pi}_{\theta_k}} \left[ \left( \frac{\pi_{\theta^{i_m}}^{i_m}(a^{i_m}|\mathrm{s})}{\pi_{\theta_k^{i_m}}^{i_m}(a^{i_m}|\mathrm{s})} - 1 \right) M^{i_{1:m}}(\mathrm{s}, \mathbf{a}) \right]$$

$$= \nabla_{\theta^{i_m}} \mathbb{E}_{\mathrm{s} \sim \rho_{\boldsymbol{\pi}_{\theta_k}}, \mathbf{a} \sim \boldsymbol{\pi}_{\theta_k}} \left[ \frac{\pi_{\theta^{i_m}}^{i_m}(a^{i_m}|\mathrm{s})}{\pi_{\theta_k^{i_m}}^{i_m}(a^{i_m}|\mathrm{s})} M^{i_{1:m}}(\mathrm{s}, \mathbf{a}) \right] - \nabla_{\theta^{i_m}} \mathbb{E}_{\mathrm{s} \sim \rho_{\boldsymbol{\pi}_{\theta_k}}, \mathbf{a} \sim \boldsymbol{\pi}_{\theta_k}} \left[ M^{i_{1:m}}(\mathrm{s}, \mathbf{a}) \right]$$

$$= \mathbb{E}_{\mathrm{s} \sim \rho_{\boldsymbol{\pi}_{\theta_k}}, \mathbf{a} \sim \boldsymbol{\pi}_{\theta_k}} \left[ \frac{\nabla_{\theta^{i_m}} \pi_{\theta^{i_m}}^{i_m}(a^{i_m}|\mathrm{s})}{\pi_{\theta_k^{i_m}}^{i_m}(a^{i_m}|\mathrm{s})} M^{i_{1:m}}(\mathrm{s}, \mathbf{a}) \right]$$

$$= \mathbb{E}_{\mathrm{s} \sim \rho_{\boldsymbol{\pi}_{\theta_k}}, \mathbf{a} \sim \boldsymbol{\pi}_{\theta_k}} \left[ \frac{\pi_{\theta^{i_m}}^{i_m}(a^{i_m}|\mathrm{s})}{\pi_{\theta_k^{i_m}}^{i_m}(a^{i_m}|\mathrm{s})} \nabla_{\theta^{i_m}} \log \pi_{\theta^{i_m}}^{i_m}(a^{i_m}|\mathrm{s}) M^{i_{1:m}}(\mathrm{s}, \mathbf{a}) \right].$$

Evaluated at $\theta^{i_m} = \theta_k^{i_m}$, the above expression equals

$$\mathbb{E}_{\mathrm{s} \sim \rho_{\boldsymbol{\pi}_{\theta_k}}, \mathbf{a} \sim \boldsymbol{\pi}_{\theta_k}} \left[ M^{i_{1:m}}(\mathrm{s}, \mathbf{a}) \nabla_{\theta^{i_m}} \log \pi_{\theta^{i_m}}^{i_m}(a^{i_m}|\mathrm{s}) \big|_{\theta^{i_m} = \theta_k^{i_m}} \right],$$

which finishes the derivation.

### D.3 PSEUDOCODE OF HATRPO

---

**Algorithm 2** HATRPO

---

1: **Input:** Stepsize $\alpha$, batch size $B$, number of: agents $n$, episodes $K$, steps per episode $T$, possible steps in line search $L$, line search acceptance threshold $\kappa$.
2: **Initialize:** Actor networks $\{\theta_0^i, \forall i \in \mathcal{N}\}$, Global V-value network $\{\phi_0\}$, Replay buffer $\mathcal{B}$
3: **for** $k = 0, 1, \ldots, K - 1$ **do**
4:     Collect a set of trajectories by running the joint policy $\boldsymbol{\pi_{\theta_k}} = (\pi_{\theta_k^1}^1, \ldots, \pi_{\theta_k^n}^n)$.
5:     Push transitions $\{(o_t^i, a_t^i, o_{t+1}^i, r_t), \forall i \in \mathcal{N}, t \in T\}$ into $\mathcal{B}$.
6:     Sample a random minibatch of $B$ transitions from $\mathcal{B}$.
7:     Compute advantage function $\hat{A}(\mathrm{s}, \mathbf{a})$ based on global V-value network with GAE.
8:     Draw a random permutation of agents $i_{1:n}$.
9:     Set $M^{i_1}(\mathrm{s}, \mathbf{a}) = \hat{A}(\mathrm{s}, \mathbf{a})$.
10:     **for** agent $i_m = i_1, \ldots, i_n$ **do**
11:         Estimate the gradient of the agent's maximisation objective
$$\hat{\boldsymbol{g}}_k^{i_m} = \frac{1}{B} \sum_{b=1}^{B} \sum_{t=1}^{T} \nabla_{\theta_k^{i_m}} \log \pi_{\theta_k^{i_m}}^{i_m} \left( a_t^{i_m} \mid o_t^{i_m} \right) M^{i_{1:m}}(s_t, \boldsymbol{a}_t).$$
Use the conjugate gradient algorithm to compute the update direction
$$\boldsymbol{x}_k^{i_m} \approx (\hat{\boldsymbol{H}}_k^{i_m})^{-1} \boldsymbol{g}_k^{i_m},$$
where $\hat{\boldsymbol{H}}_k^{i_m}$ is the Hessian of the average KL-divergence
$$\frac{1}{BT} \sum_{b=1}^{B} \sum_{t=1}^{T} \mathrm{D}_{\mathrm{KL}} \left( \pi_{\theta_k^{i_m}}^{i_m}(\cdot | o_t^{i_m}), \pi_{\theta^{i_m}}^{i_m}(\cdot | o_t^{i_m}) \right).$$
12:         Estimate the maximal step size allowing for meeting the KL-constraint
$$\hat{\beta}_k^{i_m} \approx \sqrt{\frac{2\delta}{(\hat{\boldsymbol{x}}_k^{i_m})^T \hat{\boldsymbol{H}}_k^{i_m} \hat{\boldsymbol{x}}_k^{i_m}}}.$$
13:         Update agent $i_m$'s policy by
$$\theta_{k+1}^{i_m} = \theta_k^{i_m} + \alpha^j \hat{\beta}_k^{i_m} \hat{\boldsymbol{x}}_k^{i_m},$$
where $j \in \{0, 1, \ldots, L\}$ is the smallest such $j$ which improves the sample loss by at least $\kappa \alpha^j \hat{\beta}_k^{i_m} \hat{\boldsymbol{x}}_k^{i_m} \cdot \hat{\boldsymbol{g}}_k^{i_m}$, found by the backtracking line search.
14:         Compute $M^{i_{1:m+1}}(\mathrm{s}, \mathbf{a}) = \frac{\pi_{\theta_{k+1}^{i_m}}^{i_m} \left( \mathrm{a}^{i_m} | \mathrm{o}^{i_m} \right)}{\pi_{\theta_k^{i_m}}^{i_m} \left( \mathrm{a}^{i_m} | \mathrm{o}^{i_m} \right)} M^{i_{1:m}}(\mathrm{s}_t, \mathbf{a}_t)$. //Unless $m = n$.
15:     **end for**
16:     Update V-value network by following formula:
$$\phi_{k+1} = \arg\min_\phi \frac{1}{BT} \sum_{b=1}^{B} \sum_{t=0}^{T} \left( V_\phi(s_t) - \hat{R}_t \right)^2$$
17: **end for**

---

## D.4 PSEUDOCODE OF HAPPO

---

**Algorithm 3** HAPPO

---

1: **Input:** Stepsize $\alpha$, batch size $B$, number of: agents $n$, episodes $K$, steps per episode $T$.
2: **Initialize:** Actor networks $\{\theta_0^i, \forall i \in \mathcal{N}\}$, Global V-value network $\{\phi_0\}$, Replay buffer $\mathcal{B}$
3: **for** $k = 0, 1, \ldots, K - 1$ **do**
4:     Collect a set of trajectories by running the joint policy $\boldsymbol{\pi}_{\boldsymbol{\theta}_k} = (\pi_{\theta_k^1}^1, \ldots, \pi_{\theta_k^n}^n)$.
5:     Push transitions $\{(o_t^i, a_t^i, o_{t+1}^i, r_t), \forall i \in \mathcal{N}, t \in T\}$ into $\mathcal{B}$.
6:     Sample a random minibatch of $B$ transitions from $\mathcal{B}$.
7:     Compute advantage function $\hat{A}(s, \mathbf{a})$ based on global V-value network with GAE.
8:     Draw a random permutation of agents $i_{1:n}$.
9:     Set $M^{i_1}(s, \mathbf{a}) = \hat{A}(s, \mathbf{a})$.
10:    **for** agent $i_m = i_1, \ldots, i_n$ **do**
11:        Update actor $i^m$ with $\theta_{k+1}^{i_m}$, the argmax of the PPO-Clip objective

$$\frac{1}{BT} \sum_{b=1}^{B} \sum_{t=0}^{T} \min\left( \frac{\pi_{\theta^{i_m}}^{i_m}(a_t^{i_m}|o_t^{i_m})}{\pi_{\theta_k^{i_m}}^{i_m}(a_t^{i_m}|o_t^{i_m})} M^{i_{1:m}}(s_t, \boldsymbol{a}_t),\ \text{clip}\left( \frac{\pi_{\theta^{i_m}}^{i_m}(a_t^{i_m}|o_t^{i_m})}{\pi_{\theta_k^{i_m}}^{i_m}(a_t^{i_m}|o_t^{i_m})}, 1 \pm \epsilon \right) M^{i_{1:m}}(s_t, \boldsymbol{a}_t) \right).$$

12:        Compute $M^{i_{1:m+1}}(s, \mathbf{a}) = \frac{\pi_{\theta_{k+1}^{i_m}}^{i_m}(\mathbf{a}^{i_m}|o^{i_m})}{\pi_{\theta_k^{i_m}}^{i_m}(\mathbf{a}^{i_m}|o^{i_m})} M^{i_{1:m}}(s, \mathbf{a})$. //Unless $m = n$.

13:    **end for**
14:    Update V-value network by following formula:

$$\phi_{k+1} = \arg\min_{\phi} \frac{1}{BT} \sum_{b=1}^{B} \sum_{t=0}^{T} \left( V_\phi(s_t) - \hat{R}_t \right)^2$$

15: **end for**

---

## E HYPER-PARAMETER SETTINGS FOR EXPERIMENTS

| hyperparameters | value | hyperparameters | value | hyperparameters | value |
|---|---|---|---|---|---|
| critic lr | 5e-4 | optimizer | Adam | stacked-frames | 1 |
| gamma | 0.95 | optim eps | 1e-5 | batch size | 3200 |
| gain | 0.01 | hidden layer | 1 | training threads | 32 |
| actor network | mlp | num mini-batch | 1 | rollout threads | 20 |
| hypernet embed | 64 | max grad norm | 10 | episode length | 160 |
| activation | ReLU | hidden layer dim | 64 | use huber loss | True |

Table 1: Common hyperparameters used in the SMAC domain.

| Algorithms | MAPPO | HAPPO | HATRPO |
|---|---|---|---|
| actor lr | 5e-4 | 5e-4 | / |
| ppo epoch | 5 | 5 | / |
| kl-threshold | / | / | 0.06 |
| ppo-clip | 0.2 | 0.2 | / |
| accept ratio | / | / | 0.5 |

Table 2: Different hyperparameters used for MAPPO, HAPPO and HATRPO in the SMAC

The implementation of MADDPG is adopted from the Tianshou framework (Weng et al., 2021), all hyperparameters left unchanged at the origin best-performing status.

| hyperparameters | value | hyperparameters | value | hyperparameters | value |
|---|---|---|---|---|---|
| critic lr | 5e-3 | optimizer | Adam | num mini-batch | 40 |
| gamma | 0.99 | optim eps | 1e-5 | batch size | 4000 |
| gain | 0.01 | hidden layer | 1 | training threads | 8 |
| std y coef | 0.5 | actor network | mlp | rollout threads | 4 |
| std x coef | 1 | max grad norm | 10 | episode length | 1000 |
| activation | ReLU | hidden layer dim | 64 | eval episode | 32 |

Table 3: Common hyperparameters used for IPPO, MAPPO, HAPPO, HATRPO in the Multi-Agent MuJoCo domain

| Algorithms | IPPO | MAPPO | HAPPO | HATRPO |
|---|---|---|---|---|
| actor lr | 5e-6 | 5e-6 | 5e-6 | / |
| ppo epoch | 5 | 5 | 5 | / |
| kl-threshold | / | / | / | [1e-4,1.5e-4,7e-4,1e-3] |
| ppo-clip | 0.2 | 0.2 | 0.2 | / |
| accept ratio | / | / | / | 0.5 |

Table 4: Different hyperparameters used for IPPO, MAPPO, HAPPO and HATRPO in the Multi-Agent MuJoCo domain.

| hyperparameters | value | hyperparameters | value | hyperparameters | value |
|---|---|---|---|---|---|
| actor lr | 1e-3 | optimizer | Adam | buffer size | 1e6 |
| critic lr | 1e-3 | exploration noise | 0.1 | batch size | 200 |
| gamma | 0.99 | step-per-epoch | 50000 | training num | 16 |
| tau | 5e-2 | step-per-collector | 2000 | test num | 10 |
| start-timesteps | 25000 | update-per-step | 0.05 | n-step | 1 |
| epoch | 200 | hidden-sizes | [256,256] | episode length | 1000 |

Table 5: Hyper-parameter used for MADDPG in the Multi-Agent MuJoCo domain

| task | value | task | value | task | value |
|---|---|---|---|---|---|
| Ant(2x4) | 1e-4 | Ant(4x2) | 1e-4 | Ant(8x1) | 1e-4 |
| HalfCheetah(2x3) | 1e-4 | HalfCheetah(3x2) | 1e-4 | HalfCheetah(6x1) | 1e-4 |
| Walker(2x3) | 1e-3 | Walker(3x2) | 1e-4 | Walker(6x1) | 1e-4 |
| Humanoid(17x1) | 7e-4 | Humanoid-Standup(17x1) | 1e-4 | Swimmer(10x2) | 1.5e-4 |

Table 6: Parameter kl-threshold used for HATRPO in the Multi-Agent MuJoCo domain

| hyperparameters | value | hyperparameters | value | hyperparameters | value |
|---|---|---|---|---|---|
| lr | 7e-4 | optimizer | Adam | num mini-batch | 1 |
| gamma | 0.99 | optim eps | 1e-5 | eval episode | 32 |
| gain | 0.01 | hidden layer | 1 | training threads | 1 |
| max grad norm | 10 | actor network | rnn | rollout threads | 128 |
| hidden layer dim | 64 | activation | ReLU | episode length | 25 |

Table 7: Common hyperparameters used for MAPPO, HAPPO in the Multi-Agent Particle Environment

# F   ABLATION EXPERIMENTS

In this section, we conduct ablation study to investigate the importance of two key novelties that our HATRPO introduced; they are heterogeneity of agents' parameters and the randomisation of order of agents in the sequential update scheme. We compare the performance of original HATRPO with a version that shares parameters, and with a version where the order in sequential update scheme is fixed throughout training. We run the experiments on two MAMuJoCo tasks (2-agent & 6-agent).

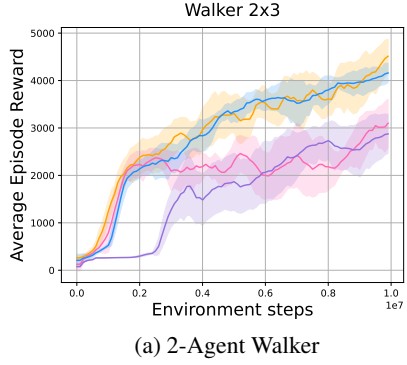
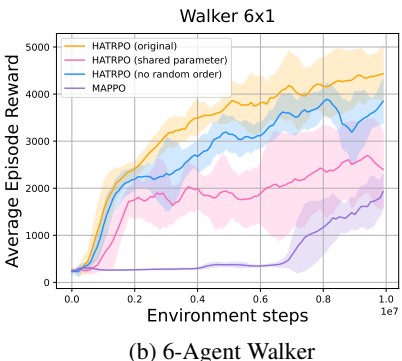

(a) 2-Agent Walker                                    (b) 6-Agent Walker

Figure 4: Performance comparison between original HATRPO, and its modified versions: HATRPO with parameter sharing, and HATRPO without randomisation of the sequential update scheme.

The experiments reveal that, although the modified versions of HATRPO still outperform baselines (represented by MAPPO), their deviation from the theory harms performance. In particular, *parameter sharing* introduces extra variance to training, harms the monotonic improvement property (Theorem 2 assumes heterogeneity), and causes HATRPO to converge to suboptimal policies. The suboptimality is more severe in the task with more agents, as suggested by Proposition 1. Similarly, *fixed order in the sequential update scheme* negatively affected the performance at convergence (especially in the task with 6 agents), as suggested by Proposition 3. We conclude that the fine performance of HATRPO relies strongly on the close connection between theory an implementation. The connection becomes increasingly important with the growing number of agents.

# G   ABLATION STUDY ON NON-PARAMETER SHARING MAPPO/IPPO

We verify that heterogeneous-agent trust region algorithms (represented by HATRPO) achieve superior performance to, originally homogeneous, IPPO/MAPPO algorithms with the parameter-sharing function switched off.

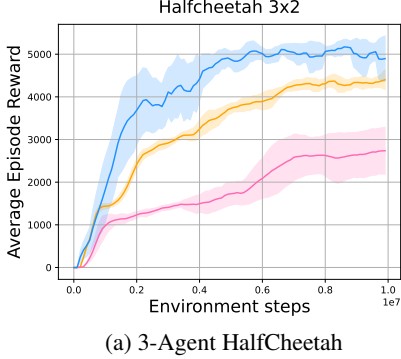
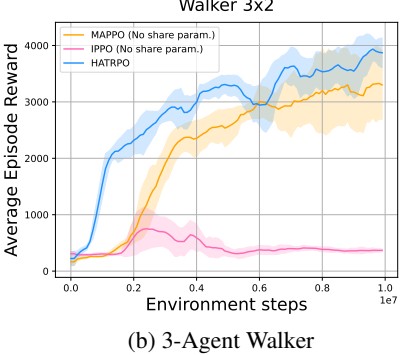

(a) 3-Agent HalfCheetah                                (b) 3-Agent Walker

Figure 5: Performance comparison between HATRPO and MAPPO/IPPO without parameter sharing. HATRPO significantly outperforms its counterparts.

# H    MULTI-AGENT PARTICLE ENVIRONMENT EXPERIMENTS

We verify that heterogeneous-agent trust region algorithms (represented here by HAPPO) quickly solve cooperative MPE tasks.

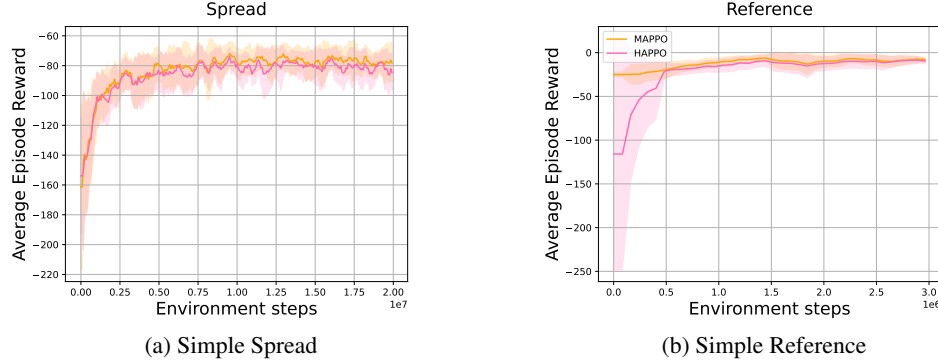

(a) Simple Spread

(b) Simple Reference

Figure 6: Performance comparison between MAPPO and HAPPO on MPE.

