# OpenReview forum: "Trust Region Policy Optimisation in Multi-Agent Reinforcement Learning"
_ICLR.cc/2022/Conference — ICLR 2022 Poster_

### Official Review · Reviewer_be2v · 2021-10-28

**Correctness:** 3
**Technical Novelty And Significance:** 3
**Empirical Novelty And Significance:** 3
**Recommendation:** 6
**Confidence:** 3

**Main Review:**

- Strengths:
	- The paper is well motivated and clearly written overall.
	- Compared to existing methods (e.g., IPPO and MAPPO), HATRPO/HAPPO can maintain the monotonic policy improvement guarantee of trust region learning in multi-agent reinforcement learning. This is a very nice theoretical property (I did not carefully check the proof though). The idea of updating each agent's policy sequentially while taking into account all previous updates is very interesting.
	- Experimental results on Multi-Agent MuJoCo look quite nice.
- Weaknesses:
	- For the experiments, I think it's important to test IPPO and MAPPO without parameter sharing to better demonstrate the advantages of the proposed methods. Also, no state-of-the-art multi-agent value-based methods are compared against.
	- On SMAC, HATRPO/HAPPO does not perform significantly better than MAPPO overall. The authors argue that it is probably because non-parameter sharing is not required in this domain, it'd then be nice to verify this hypothesis by testing MAPPO/IPPO without parameter sharing. Also, I think the results would be more impressive if the advantages of HATRPO/HAPPO over IPPO and MAPPO can be shown in another domain other than Multi-Agent MuJoCo.
- Main questions:
	- In proposition 1, why is the optimal joint reward $2^n$? Also, this means that the reward can grow without bound as the number of agents increases, what kind of (realistic) multi-agent tasks have this property?
	- The paper mentions that the proposed methods allow for flexible scheduling on the policy update order for agents at each iteration. How was this update order scheduled in the experiments?
	- On SMAC, why is IPPO and MADDPG not compared against (as in Multi-Agent MuJoCo)?
	- What is the level of partial observability (e.g., can agents see each other) in the Multi-Agent MuJoCo environments?
	- I found it a little bit surprising that HATRPO performs much better than HAPPO in most Multi-Agent MuJoCo tasks. The authors mention that it is because "the hard KL constraint in HATRPO, compared to the clipping version in HAPPO, relates more closely to Algorithm 1 that attains monotonic improvement guarantee." I don't fully understand why this makes HATRPO perform better. Can the authors comment a bit more about this?
	- The authors argue that one key advantage of HATRPO/HAPPO is that they do not need "any restrictive assumptions on decomposibility of the joint value function." I think the authors should make it clearer that this is just an advantage against most multi-agent value-based methods. I think QPLEX [1] does not place any restrictions on the decomposability of joint action-value function either. Furthermore, in a multi-agent actor-critic framework (say MAPPO), the joint value/action-value function can be decomposed in any manner too if you want to.

**Post-rebuttal:**

I would like to thank the authors for their rebuttal. It's nice to see that the authors provided some results on MAPPO/IPPO without parameter sharing. It seems that MAPPO without parameter sharing performs significantly better than MAPPO with parameter sharing. I suggest the authors to provide results on all tested Multi-Agent MuJoCo tasks, rather than on just 2 tasks in Appendix G. Overall, I think it's a good paper. I maintain my score.

[1] QPLEX: Duplex Dueling Multi-Agent Q-Learning. Wang et al.  ICLR 2021.

**Summary Of The Paper:**

This paper proposes two algorithms, HATRPO and HAPPO, which extend TRPO and PPO to cooperative multi-agent settings, respectively. Unlike existing MARL algorithms such as IPPO and MAPPO, HATRPO/HAPPO can guarantee monotonic policy improvements,  taking advantage of the proposed multi-agent advantage decomposition lemma and sequential policy update scheme. In addition, HATRPO/HAPPO can be applied to heterogeneous agents and does not require the joint value function being decomposable. Experimental results on Mulit-Agent MuJoCo demonstrate the competitive performance of the proposed methods.

**Summary Of The Review:**

I'm leaning towards accepting the paper as the proposed methods seem novel and have nice theoretical guarantee. But I'm not very confident as I believe that the paper would merit a stronger empirical section.

---

> ### Author Response · Authors · 2021-11-17
> **Response to Reviewer be2v (1/3)**
>
> ### We thank Reviewer be2v for his/her efforts in offering  constructive comments that will surely turn our paper into a better shape.
>
> 1. > **Reviewer**: The authors did not test the performance of IPPO/MAPPO without parameter sharing. They also did not evaluate any value-based method.
>
> * **Response**: Please note that MAPPO/IPPO without parameter sharing fails to work in general as we have demonstrated in a differential game with reward $r(a^1, a^2) = a^1 a^2$, illustrated on Figure 1.  However, following the reviewer's suggestion, we added results of IPPO/MAPPO without parameter sharing in the new Appendix G. We could also add results of a SOTA value-based method to our SMAC results, e.g. QMIX *[Rashid et al., 2018]*. However, we think that this is unnecessary---MAPPO, achieving 100% winning rate, is a sufficient baseline in SMAC. Furthermore, we can't, however, consider value-based methods in MAMuJoCo experiments, as the action-space of the environment is continuous.
>
> 2. > **Reviewer**: The authors claim that homogeneous policies are sufficient to solve SMAC optimally. This hypothesis needs verification through, for example, testing MAPPO/IPPO without parameter sharing. Also, the advantage of HATRPO/HAPPO in SMAC is not significant.
>
> * **Response**: We are confident that we have verified the validity of our claim by showing that (parameter-sharing) IPPO and MAPPO achieve 100% winning rate in SMAC. This finding, however, had been already made in *[Yu et al., 2021]* for all maps, and for several maps in *[Rashid et al., 2018]* and *[Peng et al., 2020]*---it is not our contribution. Because of this fact, we are not trying to claim a better performance on SMAC. Clearly, the two our algorithms perform exactly the same as MAPPO. The purpose of our SMAC experiments was to show that, just like MAPPO, our HATRPO/HAPPO achieve the optimal performance, as well as to demonstrate that the difficulty of SMAC is insufficient to discriminate between the abilities of MAPPO and our algorithms. Hence, we leaned toward MAMuJoCo, which involves complex robotic manipulations and body parts' coordination, and where sharing parameter may not be sufficient.
>
>
>
> 3. > **Reviewer**: The authors could further demonstrate the strengths of HATRPO/HAPPO by running experiments in other tasks than MAMuJoCo.
>
> * **Response**: In an alternative environment, SMAC, our methods, as well as baselines, echieve the optimal performance very quickly. We also considered a less popular MPE benchmark, but the results were the same. We added the MPE plots to the new Appendix H. Briefly speaking, the level of difficulty offered by other envionments does not allow to compare the strengths of SOTA methods. Hence, similar to the MuJoCo test bed for TRPO/PPO algorithms, we believe the Multi-Agent MuJoCo is a critical benchmark for trust-regions methods in the context of multi-agent RL.  This why we focus our experiments on mujoco tasks.
>
> 4. > **Reviewer**: Why the optimal reward in Proposition 1 is $2^n$? This is unrealistic.
>
> * **Response**: We don't claim $J^*$ to be $2^n$. The proposition states that the ratio $J^*/J^*_{\text{share}}$ equals $2/2^n$. As described in the proposition's proof in Appendix B, the optimal reward is $J^*=1$. One could simplify, and write the right-hand side of the proposition as $2^{1-n}$. If the writing is what caused confusion, we can replace $2/2^n$ with $2^{1-n}$.
>
> 5. > **Reviewer**: How was the order of updates scheduled in the experiments?
>
> * **Response**: In the experiments, we drew the order uniformly, at random, from the set of permutations of $\{1, \dots, n\}$. Randomisation of the order, which guarantees that every agent can begin the update with non-zero probability, is an important condition that lets Proposition 2 hold. Hence, our theory requires this order randomisation to assure fine performance at convergence. This has also been verified empirically through ablation experiments in Appendix F---HATRPO with order randimisation converges to better policies.
>
> 6. > **Reviewer**: Why is IPPO and MADDPG not included in SMAC experiments?
>
> * **Response**: MAPPO *[Yu et al., 2021]* is known to effectively solve all SMAC scenarios with almost all 100\% winning rate. Therefore, we consider results of IPPO and MADDPG irrelevant in terms of being the strongest baseline in SMAC.
>
> 7. > **Reviewer**: What is the level of partial observability in Multi-Agent MuJoCo?
>
> * **Response**: A robot is controlled by agents that manipulate its independent body parts. The agents can observe the global sensory input (the observation is shared across agents), but they cannot see each other's actions. This is a common setup of partial observability, also employed in previous works: *[Peng et al., 2020]*, *[Kim et al., 2021]*.

---

> ### Author Response · Authors · 2021-11-17
> **Response to Reviewer be2v (2/3)**
>
> 8. > **Reviewer**: Why is HATRPO better than HAPPO? The authors claim that is because HATRPO is more closely related to theoretically studied Algorithm 1. How does it make HATRPO better?
>
> * **Response**: HATRPO and HAPPO are deep-MARL approximations to Algorithm 1, which has fine theoretical properties: guaranteed monotonic improvement, and convergence to a Nash equilibrium. The algorithm requires agents to maximise their return surrogates $L_{\pi_k}^{i_{1:m}}(\pi_{k}^{i_{1:m-1}}, \pi^{i_m})$, but constraint the size of the update, in terms of KL-divergence, through the penalty $D_{\text{KL}}^{\max}(\pi_{k}^{i_m}, \pi^{i_m})$. HAPPO replicates this constraint by the clipping objective, which reduces the incentive of making large updates. HATRPO, however, makes sure that the size of the update is small by imposing the KL-divergence constraint on candidate policies. This hard constraint enables it to control the size of the update more firmly. As Lemma 2 revealed, the update's size must be small, to make sure that the surrogates $L_{\pi_k}$ that the agents maximise track $\mathcal{J}(\pi_{k+1})$ accurately. We believe that the supreme performance of HATRPO is due to its strictness in controlling the update size.
> We would like to remark that, although losing against HATRPO in the hardest tasks, HAPPO brings its own advantages. Its implementation is very simple, and as it uses only first-order differentiation, it is cheaper computationally.

---

> ### Author Response · Authors · 2021-11-17
> **Response to Reviewer be2v (3/3)**
>
> 9. > **Reviewer**: The authors claim that the fact that HATRPO/HAPPO manage to decompose the joint advantage (through Lemma 1) without any restrictions on the joint value function is their strength. Meanwhile, (a) there exist SOTA methods, like QPLEX, which do not make such assumptions either.
> (b) Furthermore, in actor-critic methods, any decomposition of the value function can be attained.
>
> * **Response**: (a) Indeed, to our knowledge, our multi-agent advantage theory provides the first decomposition of the value function which is rigorously consistent with the first principles of MARL. The functions that appear in Lemma 1 are the general, abstract expected values describing the policy return under particular actions. They are not defined to be outputs of neural networks that would serve agents as a proxy to guide policy updates. This stands in contrast with many (empirically great) SOTA methods, like VDN *[Sunehag et al., 2017]*, QMIX *[Rashid et al., 2018]*, or QPLEX *[Wang et al., 2020]*; each of these methods trains action-value proxies that meet the strong IGM assumption on the value function, which enables them to decompose it. These proxies, in general, are not the true value functions, and the IGM principle does not always hold.
>      * QPLEX, largely sacrifices the first principles to implement networks that bring great performance. We show how their assumptions fail in a simple  game with one state. We start by a counterexample to their Advantage-based IGM condition (Definition 1 of the QPLEX paper).
>
>          - Consider two agents $(1, 2)$ that take actions $a^1, a^2\in\{-10, -9, \dots, 9, 10\}$. They receive the reward $r(a^1, a^2) = a^1 \cdot a^2$. Every agent $i$ takes its action $a^i$ from its policy $\mu^i$ (which is an action).
>
>          - Suppose that $(\mu^1, \mu^2) = (1, -1)$.
> Under this setting, the value function is $V(s) = E[r(a^1, a^2)] = E[a^1\cdot a^2] = \mu^1\mu^2 = -1$. Now, the advantage of agent 1 is $A_1(s, a^1) = Q_1(s, a^1) - V(s) = E[r(a^1, a^2)|a^1] - \mu^1 \mu^2 = a^1\mu^2 +1 = -a^1 + 1,$ similarly the advantage of agent 2 is $A_2(s, a^2) = a^2 + 1$, and the joint advantage is $A(s, a) = Q(s, a) - V(s) = r(a^1, a^2) + 1 = a^1\cdot a^2 + 1$.
>
>           - We can see that the maximising action for agent 1 is $a_\text{max}^1 = -10$, and for agent 2 it is $a_{\max}^{2}=10$. However, the global maximisng action is either $a_{\max} = (-10, -10)$ or $a_{\max} = (10, 10)$. Hence, we have $(a^1_{\max}, a^2_{\max})\neq a_{\max}$, meaning that the advantage-based IGM does not hold in this game.
>
>      * Furthermore, in Equation (8) of QPLEX, another value function decomposability assumption is made: $V(s) = \sum_{i=1}^{n}V_i(s)$. This, again, does not hold in general. In the setting of the game described above, we have $V(s)=-1$, and $V_1(s)=Q_1(s, \mu^1)=E[Q(s, \mu^1, a^2)] = Q(s, \mu^1, \mu^2)=-1$. Similarly, $V_2(s)=-1$. Equation (8) of QPLEX would imply $-1 = -1 -1$, which we know not to hold.
>
> * (b) We are not sure what the reviewer means by *"the joint value/action-value function can be decomposed in any manner too if you want to"*. If the statement is that an arbitrary class of decentralised functions can represent $V(s)$ (or $Q(s,a)$), then this is not necessarily true in general. An example can be found in the above game example, where we showed that the decomposition $V(s) = \sum_i V_i(s)$ is not valid. It would not be valid if the game was continuous either (after replacing a deterministic policy $\mu^i$ with a continuous policy with mean $\mu^i$ the above algebra applies).
>
> ***Reference***
>
> -[Rashid et al., 2018] Rashid, Tabish, et al. "Qmix: Monotonic value function factorisation for deep multi-agent reinforcement learning." International Conference on Machine Learning. PMLR, 2018.
>
> -[Peng et al., 2020] Peng, Bei, et al. "FACMAC: Factored Multi-Agent Centralised Policy Gradients." arXiv preprint arXiv:2003.06709 (2020).
>
> -[Yu et al., 2021] Yu, Chao, et al. "The surprising effectiveness of mappo in cooperative, multi-agent games." arXiv preprint arXiv:2103.01955 (2021).
>
> -[Sunehag et al., 2017] Sunehag, Peter, et al. "Value-decomposition networks for cooperative multi-agent learning." arXiv preprint arXiv:1706.05296 (2017).
>
> -[Wang et al., 2020] Wang, Jianhao, et al. "Qplex: Duplex dueling multi-agent q-learning." arXiv preprint arXiv:2008.01062 (2020).
>
> -[Kim et al., 2021] Kim, Dong Ki, et al. "A policy gradient algorithm for learning to learn in multiagent reinforcement learning." International Conference on Machine Learning. PMLR, 2021.

---

### Official Review · Reviewer_ubwY · 2021-10-29

**Correctness:** 4
**Technical Novelty And Significance:** 3
**Empirical Novelty And Significance:** 3
**Recommendation:** 6
**Confidence:** 3

**Main Review:**

Pro:
- The paper clearly explains the related works and the improvement with respect to them. To me, the authors extend successfully a well-known state-of-the-art algorithm for single-agent RL problems in the multi-agent framework, providing potentially a new competitive algorithm also in this scenario. Claim supported by the results of the comparison with MADDPG, which is a state-of-the-art algorithm for the multi-agent problem.
- I think that giving support for heterogeneous agents is a relevant problem in order to cover more realistic problems.
- This paper provides a good number of experiments to support their claims, comparing their algorithms with the related works in many environments. Moreover, they provide the hyperparameters used in the experiments allowing results reproducibility.

Cons:
- In my opinion, it is not clearly highlighted why it is important to guarantee that the policy improves monotonically.
- The proposed algorithms work only for the cooperative setting, with a shared reward function, while MADDPG is working in both cooperative and competitive scenarios, and the letter is known to be a much harder problem. This could be potentially a weak point even if the developed algorithms show better performance in the evaluated experiments.
- The fact that the proposed approach works in cooperative scenarios only is not immediately clear neither from the title nor from the abstract and introduction. Since this represents a significant assumption, it should be made clear as soon as possible.
- The definition of Nash equilibrium for cooperative games seems quite unneeded. If all the agents share the same objective they aim at achieving the Pareto optimal solution and, given that they share the same reward, are never induced to unilaterally deviate from this solution. Can the authors explain why this definition is needed?
- The phrase "one-shot Markov game" is employed in Proposition 2, but never defined. Can the authors clarify?

Minor Issues:
- Assumption 1: this assumption is prescribing that every action is played with non-zero probability. It seems to me that this prevents from converging to deterministic policies, which can be considered a limitation of the approach. Can the authors clarify?
- Plots are not readable in grayscale, I suggest using different linestyles and/or markers
- Ticks on the plot axis are too small

**Summary Of The Paper:**

The goal of this work is to extend to the multi-agent framework the guarantee that the policy is improved monotonically after each update, as provided by TRPO for the single-agent case. The authors develop two practical algorithms where this guarantee is ensured. The algorithms are supported by a theoretical basis that explains the expansion in the multi-agent framework and relax the agents' homogeneity assumption present in the related works. An experimental evaluation showing the advantages of the proposed approach over state-of-the-art methods is provided.

**Summary Of The Review:**

Overall, although the theoretical analysis adapts well-known tools from the safe-learning literature, I think that the proposed algorithms represent an advancement for the cooperative multi-agent setting and the experimental results are able to highlight the advantages of the proposed approach.

---

> ### Author Response · Authors · 2021-11-17
> **Response to Reviewer ubwY (1/2)**
>
> ### We thank Reviewer ubwY for his/her efforts in offering  constructive comments that will surely turn our paper into a better shape.
>
> 1. > **Reviewer**: The authors did not clarify why the property of monotonic improvement is desirable.
>
> * **Response**: The justification of its importance can be found in the first paragraph of *Introduction*, where trust region methods are described in the single-agent RL framework (the only one they existed in so far). In general, a learning agent can easily make a step which degrades its performance (thus slowing down learning), or even totally breaks it down, putting the agent back at the starting point. To achieve *monotonic improvement property* means to guarantee prevention of such catastrophies, as well as to guarantee that the algorithm "works". These ascertations are even more important in MARL: a misfortunate experience of just one agent may cause the joint performance to degrade. Also, in current MARL algorithms, the agents update their independent policies without certainty that, what each agent thinks of as good, will be good for the whole team. In other words, we cannot confidently say that these methods always work. Furthermore, monotonic-improvement-driven algorithms (TRPO *[Schulman et al., 2015]* and PPO *[Schulman et al., 2017]*) were verified to achieve supreme performance with respect to non-trust-region rivals. This itself motivates the necessity for monotonic-improvement algorithms for MARL.
>
> 2. > **Reviewer**: HATRPO and HAPPO are dervied for the cooperative MARL framework (as oppose to MADDPG) which is not clearly stated from the beginning.
>
> * **Response**: Indeed, our paper assumes the cooperative MARL setting. We will make it clearer in the final version of the paper by highlighting it in *Abstract* by writing *"In this paper, we extend the theory of trust region learning to **cooperative** MARL."*, and in the title of Section 2.1. We agree with the Reviewer that extending our method to a competitive scenario is much harder, and we will leave it for future work.
>
> 3. > **Reviewer**: The notion of a Nash equilibrium is unnecessary in cooperative games because all agents who share the same objective aim at achieving the Pareto optima.
>
> * **Response**: We understand that it is tempting to think so, but Nash equilibria do exist in cooperative games, and the agents search for them. Consider a simple one-state 2-player game, where every agent can take one of two actions: $\{1, 2\}$. They receive the reward of $10$ for joint action $(1, 1)$, $9$ for joint action $(2,2)$, and $0$ otherwise. This can be represented by the following payoff matrix.
> $\begin{bmatrix}
> 10, 10 \quad \quad  0, 0\\\\
>  0, 0, \quad \quad 9, 9
> \end{bmatrix}.$
> Clearly, although the joint action $(2, 2)$ is not Pareto-optimal, it is a Nash equilibrium, and deviating from it an agent would harm its performance---the cooperativeness of the game implies that it would also harm performance of the other agent.
>
>
> 4. > **Reviewer**: The phrase "one-shot game" has not been defined.
>
> * **Response**: One-shot games are the most classical games in game theory. They refer to games that involve only one time-step, but can be equivalently represented as Markov games with one state.
> We have managed to strengthen our result, and generalise it to all cooperative Markov games. We have updated Proposition 2. Thus, the definition of one-shot games is not necessary anymore.

---

> ### Author Response · Authors · 2021-11-17
> **Response to Reviewer ubwY (2/2)**
>
> 5. > **Reviewer**: Assumption 1 insists that every action is played with non-zero probability. This prevents convergence to deterministic policies.
>
> * **Response**: Some regularity assumption (we chose $\eta$-softness for convenience) about the full coverage of every policy must be made in order for us to measure KL-divergence between any two policies, as well as to enable importance sampling (where we divide one policy by another). Indeed, in the derivations of algorithms like TRPO and PPO such an assumption was made implicitly. Furthermore, in practice, it also holds, as a neural network policy (Gaussian/softmax) never assigns zero density/mass to an action. The reviewer is right in pointing out that this prevents convergence  to deterministic policies. However,  our methods can approximate deterministic policies to any precision wanted by altering $\eta$. For our results to hold and to enable convergence to deterministic policies, we could just assume that the policies have full coverage, without the $\eta$ lower bound. Then, however, the policy space is an open set, and the existance of solutions to the problem from line 7 in Algorithm 1 needs an extra argument (but is true).---The solution always exists, as when the policy $\pi^{i_m}$ approaches the boundary of the policy space (the subset of policy space where there are actions with zero probability) the KL-penalty diverges to $\infty$. Hence, we could actually choose an $\eta_k$ (for iteration $k$) such that the potential solutions of line 7 are contained in the $\eta_k$-soft policy space. This space is compact, which implies existance of max.
>
> ***Reference***
>
> -[Schulman et al., 2015] Schulman, John, et al. "Trust region policy optimization." International conference on machine learning. PMLR, 2015.
>
> -[Schulman et al., 2017] Schulman, John, et al. "Proximal policy optimization algorithms." arXiv preprint arXiv:1707.06347 (2017).

---

> > ### Comment · Reviewer_ubwY · 2021-11-28
> > **Re: Response to Reviewer ubwY**
> >
> > I thank the authors for the answer. I appreciate that the authors have clarified in the revision that the paper focuses on cooperative MARL and, in general, for having clarified the raised issues. My evaluation was already positive, and I confirm it.

---

### Official Review · Reviewer_WgFJ · 2021-11-02

**Correctness:** 3
**Technical Novelty And Significance:** 3
**Empirical Novelty And Significance:** 3
**Recommendation:** 8
**Confidence:** 4

**Main Review:**

Strengths: This paper tackles a very important problem in cooperative MARL and provides a simple, elegant solution. The paper is strong theoretically. Unfortunately, the SMAC experiments could not distinguish the approaches compared, but the MuJoCo experiments revealed strong performance from the proposed approach.

Weaknesses:
1) The abstract says "even in cooperative games", but it's not clear that this is the primary setting until later. Cooperative MARL is not as general as MARL, so I would advise clarifying this distinction earlier on. Similarly, Markov games typically model non-cooperative games as well, i.e., each agent has their own reward function $r_i$. It would help to make this distinction clear here too (at the start of section 2 rather than right above the definition of $J(\pi)$). If "Cooperative" fits in the title, that would be great.
2) Please show the steps for arriving at $\hat{g}$ right before section 4.2. I was surprised to see the coefficient $\frac{\pi_{\theta}}{\pi_{\theta_k}} - 1$ disappear. It's fine if this goes in the appendix, but it should be derived fully somewhere. Sorry if I missed it.
3) In Figure 3, MADDPG appears to be starting to learn late in some of the experiments (e.g., d, e, f) to the point where it may surpass the performance of HATPRO which appears to asymptote. Can you include longer runs in the appendix?
4) The approach of imposing a hierarchy on players was also proposed at last years ICLR in EigenGame which solves PCA as an n-player game. The hierarchy is drawn randomly at the beginning of the game and held fixed for all time whereas you draw a new one at every iteration. It was also critical there for (proving) and accelerating convergence to the equilibrium. Maybe this is a more general principle for n-player games that is worth discussing in your paper.
5) I am also curious to see a discussion of the optimal hierarchy and what performance might be lost by randomly sample a hierarchy uniformly at every iteration.

Minor:
- Choice of "sequential": it was not clear to me initially that this referred to the update being sequential across the players. Maybe the word "hierarchical" or being more explicit about what is meant by sequential could help.
- Definition 1: Should you add $m < n$?
- I found it a bit confusing to use $L$ for something we want to maximize. Why not $f, g, U$?
- Theorem 1: The expectation $\mathbb{E}_{s \sim \rho_\pi}$ is defined w.r.t. an improper distribution. Can you be more rigorous (proper) here?
- Above definition 2: "assuming agent $i_1$ takes an action $\bar{a}^{i_m}$ such that $A^{i_m}(\ldots) > 0$". Can you comment on feasibility here? I expect it is always possible to find $A \ge 0$. When only $A=0$ is available for all agents, then the agents have reached a local max. Otherwise, there is room for improvement. Can you add a discussion like that if that is correct?
- Definition 2: You say let "$\bar{\pi}^{i_{1:m-1}}$ be some other joint policy". I guess the emphasis should be on "other" here. Can you not explicitly define $\pi = \prod_{n=1}^N \pi^{i_n}$ and $\bar{\pi}^{i_{1:m-1}} = \prod_{n=1}^{m-1} \hat{\pi}^{i_n}$? The details here are important and it was a bit hard to visually follow all the subscripts. Any help you can give the reader would be appreciated.
- Given this is a cooperative MARL paper, I found the Nash result interesting, but unnecessary and possibly a distraction. I suggest it can be moved to the appendix if you need space.
- Please define "one-shot" games. Are these not just multi-armed bandits? Looking at the appendix, under equation (17) on p. 20, it seems you should replace $s_t$ with $s$ or explicitly define a "one-state" game as $s_t = s \forall t$.
- typo "vaires" at the bottom of page 6

**Summary Of The Paper:**

This paper introduces a trust region policy optimization method for cooperative multiagent reinforcement learning with a improvement guarantee. This is accomplished by imposing an arbitrary hierarchy on the players so that each player optimizes its policy w.r.t. the players who updated before it in the hierarchy. To my knowledge, this is the first extension of PPO and TRPO to the MARL setting with an improvement guarantee. Experiments demonstrate the efficacy of the approach.

**Summary Of The Review:**

In my opinion, this is an important theoretical result for cooperative MARL with good empirical support from experiments.

---

> ### Author Response · Authors · 2021-11-17
> **Response to Reviewer WgFJ**
>
> ### We thank Reviewer WgFJ for his/her efforts in offering constructive comments that will surely turn our paper into a better shape.
>
> 1. > **Reviewer**: The paper deals with the cooperative MARL framework, which is not clearly communicated at its beginning.
>
> * **Response**: We apologise for this mistake from our side. We will change the sentence from abstract *"In this paper, we extend the theory of trust-region learning to MARL."* to *"In this paper, we extend the theory of trust-region learning to cooperative MARL."*, as well as the title of Section 2.1. We are also considering following the Reviewer's suggestion to put "cooperative" in the title.
>
> 2. > **Reviewer**: The derivation of $\hat{g}^{i_m}$ is missing. In particular, lack of the coefficient $\pi_{\theta^{i_m}}^{i_m}(a^{i_m}|s)/\pi_{\theta^{i_m}_k}^{i_m}(a^{i_m}|s)-1$ is confusing.
>
> * **Response**: We acknowledge we omitted the derivation of $\hat{g}^{i_m}$, as its form is a natural consequence of the policy gradient theorem. We recognise, however, that the lack of "-1" in the term involving $\pi_{\theta^{i_m}}^{i_m}$ is confusing. To clarify, "-1" disappears after Equation (10) as it only introduces a constant, with zero gradient, to the optimisation objective of agent $i_m$. This move was also a subject of our discussion: some argued that including the explanation may cause more confusion. We now see that the result is opposite---we have put the explanation back by saying *"the term $-1\cdot M^{i_{1:m}}(s, a)$ is not reflected in $\hat{g}^{im}$, as it only introduces a constant with zero gradient"*. We have also provided the full derivation of the gradient in the new Appendix D.2.
>
>
> 3. > **Reviewer**: In an ICLR 2021 paper on *EigenGame*, the approach of a hierarchy on the player update was described. Similarly to the case of HATRPO, scheduling of the hierarchy served as a tool for proving a convergence to Nash. Authors may consider discussing this similarity.
>
> * **Response**: We thank the reviewer for pointing out the potential connection to the work of EigenGame in terms of scheduling the hierarchy. We would like to point out that our sequential update scheme is different from that of EigenGame, where the players perform a sequence of complete, individual learning processes---in HATRPO/HAPPO, the joint learning process consists of iterations involving sequential updates. Attempting to train every agent one-by-one is not the purpose of this work, and seems impractical in deep MARL. To see this, notice that the optimal policy, that the first agent would learn, would be the best response policy to randomly initialised (thus unreasonable) policies of other agents. Such a policy would not be useful, thus giving an end to the connection. Furthermore, the update rules in the two papers are clearly different; EigenGame algorithms take advantage of matrix calculus identities, while HATRPO/HAPPO leverage Markov game theorems (e.g., Lemmas 1 & 2 of our paper). We added a relevant piece of discussion in *Related Workds*.
>
>
> 4. > **Reviewer**: It would be interesting to see the discussion about the choice and usage of the optimal hierarchy, and what advantage it brings.
>
> * **Response**: This comment opens an interesting question of optimality. However, in our algorithms there is no optimal hierarchy. Indeed, Proposition 2 requires that every agent has a lower-bounded probability of opening the update---fixing a hierarchy would violate it.
>
> 5. > **Reviewer**: A list of "minor" issues.
>
> * **Response**: We carefully revised the list of nine points that need clarification, or more precision. We are confident about the correctness of our paper against some of the concerns. Here, we discuss the ones that we think of as most significant.
>     * **Point 5.** We agree that finding an action with positive advantage is not always possible. However, the role of the paragraph is not to state a theorem or introduce an algorithm---it is to prvide an intution behind a possible application of Lemma 1, which is learning with monotonic improvement. We believe that the strictness here makes this intution clearer.
>     * **Point 6.** We understand the reading difficulty that our notation caused. We like the idea of introducing different policies proposed by the reviewer, and have incorporated them in our paper. We also rewrote Proposition 3 with this concept.
>     * **Points 7 & 8.** We have managed to strengthen our results from one-shot games (games with one state) to Markov games. The new result replaced the old one in our paper.

---

> > ### Comment · Reviewer_WgFJ · 2021-11-19
> > **Thank you for clarifying**
> >
> > Thank you for these clarifications. I maintain my score: (8) accept, good paper.
> >
> > Regarding the hierarchy, thank you for pointing out fixing the hierarchy violates Proposition 2. It still seems you have the freedom the skew this distribution towards some hierarchies, no? In an environment where agents have very different skills, I'm curious. Do you think this distribution over permutations (hierarchies) could be meta-optimized to accelerate learning?

---

> > > ### Author Response · Authors · 2021-12-01
> > > **Meta-learning of the update order**
> > >
> > > Thank you for proposing this interesting idea. Although we have not managed to develop and test a method that would assure an affirmative answer to this question, we have made some conceptual progress. Below, we describe the results of our efforts.
> > >
> > > We tried to formulate a meta-learning objective of which solution guarantees better convergence properties (in terms of the limit equilibria), but this may not make sense, even in highly heterogeneous games, due to the following reason: if the Nash equilibrium is unique, all such update designs give the same solution, by Proposition 2.
> > >
> > > As the Reviewer suggested, one can try to optimize for the learning speed. This might be possible to learn, even in parallel with the training of agents. Consider a heuristic, in which we compute some score for each agent, which describes what "opportunity" for improvement it has; intuitively if an agent puts a very small weight $\pi(a|s)$ on some action with a large advantage $A(s, a)$, and large weight on actions with very negative advantage, it has a large chance for improvement. We could sort the agents according to that score, and design a (lower-bounded, for Proposition 2 guarantees) distribution that favors agents with the large scores. In this way, we can guarantee that the updates due to the first agents in the sequence are substantial.
> > >
> > > If we manage to develop a method that would optimize the order distribution, we will be happy to present its performance in the appendix of the final version of the paper.
> > >
> > > Authors

---

### Official Review · Reviewer_ZAQm · 2021-11-03

**Correctness:** 3
**Technical Novelty And Significance:** 3
**Empirical Novelty And Significance:** 3
**Recommendation:** 6
**Confidence:** 4

**Main Review:**

The problem is well-formulated, but difficult to follow in some sections. Experiments are validating the claims authors made in the paper. I have the following concerns and questions regarding this paper. I can adjust my score accordingly after the authors reply to these questions:

(1) I have a bit of difficulty understanding Lemma 2 intuitively. Is there any guarantee on the positive update to happen? What if $L_\pi^{i_{1:m}}(.,.)-CD_{\mathrm{KL}}^{{\mathrm{max}}}(\pi^{i_m}_k,\pi^{i_m})$ is negative? In other words, is the  joint policy improvement guarantee based on a positivity assumption?

(2) Algorithm 1, line 7, based on my understanding from Lemma 2, inside the argmax the compliment sign should be used as: $CD_{\mathrm{KL}}^{\mathrm{max}} (.,\pi^{-i_m})$
not
 $CD_{\mathrm{KL}}^{{\mathrm{max}}}(.,\pi^{i_m})$. Could you correct me if I am wrong?

(3) Some intuitions are required before/after Proposition 2 to let the reader know the outcome of this proposition. In fact, based on Appendix C.3, some more auxiliary definitions are required to prove the Nash equilibrium. Also I have some doubts regarding the one-shot Markov game definition, could you elaborate?

(4) Doesn’t proposition 3 violate the assumption of HATPRO on “agents do not share parameters” and “there is no assumption on the  decomposability of the joint value function”?

(5) There are many grammatical mistakes in this paper that require a thorough proofread. E.g.
Permutaion, theorm, enviornment,...


**Summary Of The Paper:**

This paper proposes Heterogeneous-Agent Trust Region Policy Optimisation (HATPRO), in which: (1) agents do not share parameters (2) there is no assumption on the  decomposability of the joint value function. Two adaptive algorithms are introduced to enable sequential updating that benefit from a monotonic improvement guarantee.

**Summary Of The Review:**

The paper is strong in theoretical presentation and experiments are sufficient. However, the readability of the paper is an issue and requires some modifications.

---

> ### Author Response · Authors · 2021-11-17
> **Response to Reviewer ZAQm (1/2)**
>
> ### We thank Reviewer ZAQm for his/her efforts in offering constructive comments that will surely turn our paper into a better shape.
>
> 1. > **Reviewer**: The claim about the improvement of the joint policy relies on the statement that $
> \max_{\pi^{i_m}}L_{\pi_k}^{i_{1:m}}(\pi_{k+1}^{i_{1:m-1}}, \pi^{i_m}) - CD_{\text{KL}}^{\max}(\pi_k^{i_m}, \pi^{i_m})
> $
> is not negative. Is this an assumption, or is it guaranteed?
>
> * **Response**: Although we understand this might be surprising, this result is indeed guaranteed. The pieces explaining this phenomenon are collected in the proof of Theorem 2 (Appendix C.2, below Equation (15)), and we give the following sketch of the proof.
>     * If the expression is maximised over all $\pi^{i_m}$, in particular, the max is greater than the expression evaluated at $
> \pi^{i_m}=\pi_{k}^{i_m}$. We have
> $\max_{\pi^{i_m}} L_{\pi_k}^{i_{1:m}}(\pi_{k+1}^{i_{1:m-1}}, \pi^{i_m}) - CD_{\text{KL}}^{\max}(\pi_{k}^{i_m}, \pi^{i_m}) \geq L_{\pi_k}^{i_{1:m}}(\pi_{k+1}^{i_{1:m-1}}, \pi_{k}^{i_m}) - CD_{\text{KL}}^{\text{max}}(\pi_{k}^{i_m}, \pi_{k}^{i_m}).$
>     * By Equation (6) (in the remark under Definition 2), we know that $L_{\pi_k}^{i_{1:m}}(\pi_{k+1}^{i_{1:m-1}}, \pi^{i_m}_k) = 0$.
>     * The KL divergence between a policy and itself is zero: $CD_{\text{KL}}^{\max}(\pi_k ,\pi_k) = 0$.
>     * Altogether, $L_{\pi_k}^{i_{1:m}}(\pi_{k+1}^{i_{1:m-1}}, \pi_k^{i_m}) - CD_{\text{KL}}^{\max}(\pi_k^{i_m}, \pi_k^{i_m})=0.$
>
>    Hence, there is a guarantee on the expression raised in the question being non-negative. As the sum (over all agents) of these expressions provides a (non-negative) lower bound on the difference $\mathcal{J}(\boldsymbol{\pi}_{k+1}) - \mathcal{J}(\boldsymbol{\pi}_k)$ (Lemma 2), the joint return is guaranteed not to decrease.
>
> 2. > **Reviewer**: There might be a typo in Algorithm 1: in line 7, the penalty term should be written as $-CD_{\text{KL}}^{\max}(\pi_{k}^{i_m} \pi^{-i_m})$, rather than $-CD_{\text{KL}}^{\max}(\pi_{k}^{i_m}, \pi^{i_m})$. Is that right?
>
> * **Response**: This is not a typo, nor an error. The KL-divergence term $D_{\text{KL}}^{\max}(\pi_{k}^{i_m}, \pi^{i_m})$ puts a penalty on *candidate policies* $\pi^{i_m}$ of agent $i_m$, that are very distanced (in KL-divergence) from the *current policy* $\pi_{k}^{i_m}$ of agent $i_m$.
> Similarly, the terms that appeared in Lemma 2 were $D_{\text{KL}}^{\max}(\pi^{i_m}, \bar{\pi}^{i_m})$, where the second argument is "**pi-bar**"$^{i_m}$. In this Lemma, $\pi^{i_m}$ was denoting the current policy, while $\bar{\pi}^{i_m}$ was a candidate (see the notation statement under Definition 1). As far as we understand, this was confused with the joint policy of other agents $\boldsymbol{\pi}^{-i_m}$, because the "bar" over $\pi$ looks similarly to a "minus". To make it clearer, we are also thinking to replace $\max_{\pi^{i_m}}$ in line 7 of Algorithm 1 with $\max_{\bar{\pi}^{i_m}}$ (max over pi-bar$^{i_m}$), for consistency with the notation statement.
>
> 3. > **Reviewer**: The authors should provide more details, perhaps in an intuitive form, behind the proof of Proposition 2. Also, it is not exactly clear what are the "one-shot games" the authors refer to in the proposition's claim.
>
> * **Response**: Firstly, we would like to clarify the setting: a *one-shot game* is classical game-theoretic framework of a game which is played only once (it ends once the agents have taken their actions). Equivalently, it can be represented as a Markov game with only one state.
> The reviewer made us realise that introducing the result about convergence to Nash (Proposition 2), without any further explanation, might seem unintuitive. Following the Reviewer's suggestion, we have provided a clarification about the connection between the algorithm's steps and their application to the proof, under the proposition. Furthermore, we are happy to announce that we managed to strengthen the result, and prove the Nash **convergence of Algorithm 1 in cooperative Markov games** (with many states). The new result replaced the claim about one-shot games in our paper.
> However, we still believe putting all the auxiliary definitions from Appendix C.3 in the main paper is unnecessary.

---

> > ### Comment · Reviewer_ZAQm · 2021-12-01
> > **Thanks for clarification!**
> >
> > I appreciate your response and I checked the updated manuscript. I will keep my current score.

---

> ### Author Response · Authors · 2021-11-17
> **Response to Reviewer ZAQm (2/2)**
>
> 4. > **Reviewer**: Does not Proposition 3 (a) violate the assumptions on the heterogeneity of agents? (b) assume decomposability of the joint value function?
>
> * **Response**: (a) The proposition is derived for the general heterogeneous setting. It describes how an expectation over candidate policie**s** of the multi-agent advantage $A_{\boldsymbol{\pi}}^{i_m}$ can be estimated by sampling from the data collected with old policies. As far as we understand, the reviewer is concerned about the advantage $A_{\boldsymbol{\pi}}$ being joint. This "jointness", however, comes from the fact that we work in the cooperative framework, where the reward is joint (see Section 2.1). The joint policy $\boldsymbol{\pi}$, which the advantage is built upon, is a product of heterogeneous policies, $\boldsymbol{\pi}(\cdot|s) = \prod_{i=1}^{n}\pi^i(\cdot^i|s)$.
> (b) Proposition 3 does not assume the decomposability of the value function. Indeed, its statement and proof involve the decomposition of advantage---this, however, is achieved from the first principles by leveraging Lemma 1. The lemma, without making any assumptions on the value function (e.g. VDN *[Sunehag et al., 2017]*/QMIX *[Rashid et al., 2018]*), reveals how to decompose the joint advantage into a sum of (unfolding) multi-agent advantages; it is **one of the key novelties** of the paper, which we would like to highlight.
>
> ***Reference***
>
> -[Sunehag et al., 2017] Sunehag, Peter, et al. "Value-decomposition networks for cooperative multi-agent learning." arXiv preprint arXiv:1706.05296 (2017).
>
> -[Rashid et al., 2018] Rashid, Tabish, et al. "Qmix: Monotonic value function factorisation for deep multi-agent reinforcement learning." International Conference on Machine Learning. PMLR, 2018.

---

### Author Response · Authors · 2021-11-17
**Summary of responses and major amendmends**

Dear Reviewers,

Thank you for your thorough investigation of our paper, and for the insightful comments. We have carefully read them, and in response, we did our best to resolve any confusion caused by our writing. Furthermore, we considered potential improvements to our work, which we included in the updated submission version. Here, we summarise the amendments.

- We made it clear that our paper works in the cooperative MARL framework by highlighting it in Abstract, and in the title of Section 2.1.
- We included the missing discussion on the connection to EigenGame in the Related Work section.
- We clarified our notation in Definition 2.
- We provided the details of the HATRPO gradient derivation in Appendix D.2.
- We provided ablation studies to baseline algorithms (IPPO and MAPPO) in Appendix F, and experiments on other environments (MPE) in Appendix H.
- We extended the result of Proposition 2, from one-shot games (games with one state) to cooperative Markov games (games with many states). We updated the claim and its proof accordingly.

We hope that these changes answer the Reviewers' concerns, and make our paper more readable and insightful.

Authors

---

### Public Comment · ~Rongye_Shi1 · 2022-12-12
**Request a clearification on why supplementary eq.(17) is equivalent to Bellman optimality equation**

Dear Authors,

This paper meaningfully contributes to extending TRPO to MARL, and I enjoy reading this paper. However, I have a question regarding why supplementary eq.(17) is equivalent to (or imply) Bellman optimality equation, as stated in Append C.3, "Step 3 (optimality)".

Specifically, this statement is true only if the TR-stationary policy $\bar{\pi}^i$ for agent $i$ can be a greedy policy, which is contradictory to the "Assumption 1" in your main text (i.e., the policy space $\Pi^i$ is $\eta$-soft with $0<\eta \leq 1$).

It is true that the greedy policy $\bar{\pi}^i(a^i|s)=max_{\hat{a}^i}Q^i_{\bar{\pi}}(s, \hat{a}^i)$ is the solution of

$$\bar{\pi}^i= \underset{\pi^i}{\mathrm{argmax}}E_{a^i\sim \pi^i}[A^i_{\bar{\pi}}(s,a^i)]=\underset{\pi^i}{\mathrm{argmax}}E_{a^i\sim \pi^i}[Q^i_{\bar{\pi}}(s,a^i)],$$

making the TR-stationary policy $\bar{\pi}^i$ a solution of the Bellman optimality equation. However, I am confused by how this greedy policy is consistent to the $\eta$-soft assumption. Thank you.

---

> ### Public Comment · ~Jakub_Grudzien_Kuba1 · 2022-12-13
> **Clarification**
>
> Dear Rongye,
>
> Thank you for your kind words. Your question shows us that you read the paper in detail, which brings us extra satisfaction. I understand your confusion and will do my best to clarify it.
>
> If you look at Step 3 again you will notice that we never said that the policy $\bar{\pi}^i$ is greedy. That is because of Assumption 1 that you have cleverly connotated. The point we make is that this Bellman optimality equation holds with respect to the $\eta$-soft policy space, and thus implies optimality of $\bar{\pi}^i$ in $\eta$-soft policy space. Any proof of policy optimality under this equation is valid. I will leave one (equivalently in this case, for single RL) at the bottom of this message. Your question shows us that we should have made it more clear in the theorems' statements that the results hold for our choice of policy space. I will keep that in mind for the future. Have things become more clear now?
>
> As an aside, I will add that while the $\eta$-softness assumption was convenient, it seems to me (I speculate) that it is not necessary.
>
> Best wishes,
> Kuba
>
> Proof:
> Suppose $\bar{\pi}$ satisfies the Bellman optimality equation with respect to the $\eta$-soft policy space. Suppose, for contradiction, that there exists an $\eta$-soft policy $\pi_{+}$ whose expected return is strictly higher than that of $\bar{\pi}$. Then, by *performance difference lemma* (Lemma 7) applied to $J(\pi_{+})-J(\bar{\pi})$, there must exist a state $s$ such that $E_{a\sim\pi_{+}}[A_{\bar{\pi}}(s, a)] > E_{a\sim\bar{\pi}}[A_{\bar{\pi}}(s, a)]$, which is a contradiction with Bellman optimality equation. This finishes the proof.

---

> > ### Public Comment · ~Rongye_Shi1 · 2022-12-17
> > **let's look into the Bellman optimality equation**
> >
> > Thank you for the detailed reply. I think my question could be rephrased as whether a policy in the $\eta$-soft space $\Pi$ can satisfy the Bellman optimality equation (BOE).
> >
> > Per my understanding on the BOE definition, if we define $V^*(s)= \underset{\pi \in \Pi}{\mathrm{max}}V_{\pi}(s)$ and define $Q^*(s,a)= \underset{\pi \in \Pi}{\mathrm{max}}Q_{\pi}(s,a)$, the BOE says that the following equation should hold: $V^*(s)=\underset{a \in A}{\mathrm{max}}Q^*(s,a), \forall s \in S$, i.e., $V_{\pi^*}(s)=\underset{a \in A}{\mathrm{max}}Q_{\pi^*}(s,a)$ when ${\pi^*}$ is an optimal policy.
> >
> > However, if ${\pi^*}$ is $\eta$-soft, the actions other than the optimal one $a^*=\underset{a \in A}{\mathrm{argmax}}Q_{\pi^*}(s,a)$ has at least $\eta$ probability to be taken. As a result $V_{\pi^*}(s)=\sum_{a \in A} \pi^*(a|s)Q_{\pi^*}(s,a)$ might not be equal to $\underset{a \in A}{\mathrm{max}}Q_{\pi^*}(s,a)$ if there exists at least one bad action $a'$ to make the $Q_{\pi^*}(s,a')$ smaller than $\underset{a \in A}{\mathrm{max}}Q_{\pi^*}(s,a)$.
> >
> > The only option to have the BOE hold in the $\eta$-soft space seems to make any action to be the optimal one, i.e., $Q_{\pi^*}(s,a')=\underset{a \in A}{\mathrm{max}}Q_{\pi^*}(s,a), \forall a' \in A$, which is not the one we want to achieve.
> >
> > Pardon me if I have any misunderstanding. If so, please correct me. Thank you.

---

> > > ### Public Comment · ~Jakub_Grudzien_Kuba1 · 2022-12-17
> > > **Further clarification**
> > >
> > > Hi Rongye,
> > >
> > > Thank you for a careful explanation of your thoughts.
> > >
> > > I am not sure if my point was clearly made. The optimal policy in our paper is an optimal $\eta$-soft policy, which is not necessarily the "generally" optimal one that you seem to be referring to. Hence, your efforts to analyze how our proof relates to optimal greedy policies may not be necessary.
> > >
> > > Nevertheless, I think you make the right point. I just had a browse and realized that I had misused the term "Bellman optimality equation". Certainly, I do not claim to have $V_{\pi}(s) = Q_{\pi}(s, a^{*})$. The first equation under the Step 3 part of the proof is just an optimality criterion for a policy that does not seem to have a name. My apologies for the confusion. I hope that you are familiar with that criterion, or the short proof I stated above explains it clearly.  Again, I appreciate reading the paper in detail.
> > >
> > > Kuba

---

> > > > ### Public Comment · ~Rongye_Shi1 · 2022-12-18
> > > > **Thank you for your clarification**
> > > >
> > > > Thank you Kuba for your clarification. I'm good with your short proof. No further issue on my side now. Seems like the term "Bellman optimality equation" is also misused in your other papers such as the ICML 2022 (Mirror Learning), referring to the optimality criterion. Maybe, it would be better to coin a technical name for the introduced optimality criterion in your future papers to avoid confusion.
> > > >
> > > > Best,
> > > > Rongye

---

> > > > > ### Public Comment · ~Jakub_Grudzien_Kuba1 · 2022-12-19
> > > > > **Cheers**
> > > > >
> > > > > Haha, indeed, I should be more careful with terminology. I'm quite sloppy with names - I devised the term Mirror Learning too, and while it is completely different, people confuse it with Mirror Descent. Thank you for your advice and for a fruitful discussion.
> > > > >
> > > > > Kuba

---

### Decision · Program_Chairs · 2022-01-20

**Decision:**

Accept (Poster)

**Comment:**

The submission proposes a new approach to deriving a policy gradient type algorithm for multi agent RL (MARL) where the agents are interested in a common objective but with potentially different action spaces. It extends the monotone improvement property for single agent trust-region based methods like TRPO to a multi agent update setting where the updates are performed in sequence by the agents, and uses this idea to derive new multi agent analogues of TRPO and PPO. These algorithms are shown to be competitive with existing strategies for MARL on a Starcraft environment, and superior in the case of common Mujuco benchmarks.

All reviewers are unanimous in their appreciation for the paper's contributions. The initial concerns about clarity of the technical results, especially the improvement guarantee of the key lemma, that some reviewers had were addressed adequately by the author responses. Hence, I gladly recommend acceptance.